# A virus-packageable CRISPR screen identifies host factors mediating interferon inhibition of HIV

**Molly OhAinle[1]\*, Louisa Helms[1], Jolien Vermeire[1], Ferdinand Roesch[1], Daryl Humes[1], Ryan Basom[2], Jeffrey J Delrow[2], Julie Overbaugh[1], Michael Emerman[1]\***

[1]Divisions of Human Biology and Basic Sciences, Fred Hutchinson Cancer Research Center, Washington, United States; [2]Genomics and Bioinformatics Shared Resource, Fred Hutchinson Cancer Research Center, Seattle, United States

**Abstract** Interferon (IFN) inhibits HIV replication by inducing antiviral effectors. To comprehensively identify IFN-induced HIV restriction factors, we assembled a CRISPR sgRNA library of Interferon Stimulated Genes (ISGs) into a modified lentiviral vector that allows for packaging of sgRNA-encoding genomes *in trans* into budding HIV-1 particles. We observed that knockout of Zinc Antiviral Protein (ZAP) improved the performance of the screen due to ZAP-mediated inhibition of the vector. A small panel of IFN-induced HIV restriction factors, including MxB, IFITM1, Tetherin/BST2 and TRIM5alpha together explain the inhibitory effects of IFN on the CXCR4-tropic HIV-1 strain, HIV-1$_{LAI}$, in THP-1 cells. A second screen with a CCR5-tropic primary strain, HIV-1$_{Q23.BG505}$, described an overlapping, but non-identical, panel of restriction factors. Further, this screen also identifies HIV dependency factors. The ability of IFN-induced restriction factors to inhibit HIV strains to replicate in human cells suggests that these human restriction factors are incompletely antagonized.

**Editorial note:** This article has been through an editorial process in which the authors decide how to respond to the issues raised during peer review. The Reviewing Editor's assessment is that all the issues have been addressed (see decision letter).

DOI: https://doi.org/10.7554/eLife.39823.001

**\*For correspondence:** mohainle@fredhutch.org (MOA); memerman@fredhutch.org (ME)

**Competing interests:** The authors declare that no competing interests exist.

## Introduction

The HIV-1 pandemic resulted from a series of successive cross-species transmissions of primate lentiviruses. Simian Immunodeficiency Virus (SIV) transmission from African Old World primates to chimpanzees yielded the recombinant virus SIVcpz, which ultimately crossed into humans (*Sharp and Hahn, 2011*). Successful replication of lentiviruses in a new host species required adaptation to restriction factors in the new host (*Etienne et al., 2015*; *Etienne et al., 2013*; *Kirmaier et al., 2010*). Restriction factors that target primate lentiviruses include TRIM5alpha, MxB, Tetherin, SAMHD1, the APOBEC3 family of cytidine deaminases (*Malim and Bieniasz, 2012*) and more recently described factors such as SERINC3/5, Zinc Antiviral Protein (ZAP), GBP5, SLFN11, LGALS3BP (90K), the HUSH complex (*Chougui et al., 2018*; *Krapp et al., 2016*; *Li et al., 2012*; *Lodermeyer et al., 2013*; *Rosa et al., 2015*; *Takata et al., 2017*; *Usami et al., 2015*; *Yurkovetskiy et al., 2018*), as well as nearly 200 other proposed factors (reviewed in (*Gélinas et al., 2018*). HIV-1 has evolved accessory proteins that degrade many host restriction factors (*Duggal and Emerman, 2012*). Further, mutations preventing recognition by restriction factors, such as evolution of low CG dinucleotide content in the HIV-1 genome (*Takata et al., 2017*) or mutations in capsid (*Kirmaier et al., 2010*; *Wu et al., 2013*), represent another mechanism of escape.

**eLife digest** The different strains of the human immunodeficiency virus (or HIV) can infect a variety of cells in the human body. When a cell senses being attacked, it can defend itself using molecules called restriction factors, which are created under the control of a signal known as interferon.

Researchers have already identified several restriction factors, using techniques that are relatively laborious and time-consuming, but many questions remain about these proteins. Here, Ohainle et al. created a new method to screen for restriction factors; by harnessing the CRISPR/Cas9 technique, HIV was tricked into revealing its own weaknesses. The method allowed Ohainle et al. to make precise, targeted changes to thousands of genes that are turned on by interferon, and deactivate them. The experiments revealed that HIV multiplied better in human cells in which several specific genes had been neutralized. This suggests that these genes encode restriction factors that are activated by interferon to combat HIV. The combined action of a few of these proteins can fight the virus, even if it cannot completely eradicate it. Further experiments found that a different, but overlapping set of restriction factors defended cells against a different strain of HIV.

The method developed by Ohainle et al. is a useful tool to identify new restriction factors. By dissecting the role of these proteins in keeping different HIV strains under control, we may understand how the virus has become dangerous for humans by evading some of these defenses. Ultimately, this could help with finding better ways to fight this deadly disease.
DOI: https://doi.org/10.7554/eLife.39823.002

Many restriction factors that target HIV-1 are induced by type I Interferon (IFN) and are therefore Interferon-Stimulated Genes (ISGs). Interferon has been implicated in at least partial control of HIV replication in chronically-infected individuals treated with IFN (*Asmuth et al., 2010*; *Azzoni et al., 2013*) as well as in SIV-infected rhesus macaques (*Sandler et al., 2014*). In contrast, IFN levels have also been correlated with higher viral load and decreased CD4 T cell counts in HIV-infected individuals (*Hardy et al., 2013*). Further, it appears that ISG expression exerts changing selective pressure on HIV evolution in vivo since transmitted/founder (T/F) strains are relatively resistant to IFN compared to viruses isolated later in infection (*Fenton-May et al., 2013*; *Iyer et al., 2017*; *Parrish et al., 2013*). It remains to be determined if one dominant ISG mediates all or most of the IFN inhibition, or if a multitude of antiviral ISGs together limit viral replication in response to IFN.

The HIV-1 LAI strain (HIV-1$_{LAI}$) was isolated from a chronically-infected individual (*Wain-Hobson et al., 1991*) and is sensitive to type I IFN. Specifically, potent IFNα inhibition of HIV-1$_{LAI}$ can be observed in the THP-1 monocytic cell line (*Goujon and Malim, 2010*). MxB, an interferon-induced GTPase that binds to and blocks lentiviral capsids, was identified as an IFNα-induced factor in THP-1 cells (*Goujon et al., 2013*; *Haller et al., 2015*; *Kane et al., 2013*; *Liu et al., 2013*), although the role of MxB in the IFNα-induced inhibition of HIV infection in these cells has been questioned (*Opp et al., 2015*). Restriction factors have previously been discovered through cDNA library screening or by comparing expression of transcript levels in permissive versus non-permissive cells (*Goujon et al., 2013*; *Kane et al., 2013*; *Neil et al., 2008*; *Sheehy et al., 2002*; *Stremlau et al., 2004*). More high-throughput approaches to find HIV restriction factors have focused on either over-expression screens to identify broad antiviral ISGs (*Schoggins et al., 2011*) or HIV-specific antiviral ISGs (*Kane et al., 2016*). Further, one screen for HIV restriction factors was also performed by transfection of siRNA pools (*Liu et al., 2011*). However, a more complete understanding of the constellation of restriction factors that inhibit HIV in human cells and a more tractable, high-throughput method to discover restriction factors remains to be described.

Here we describe a CRISPR/Cas9-mediated gene knockout functional screening method in which lentiviral genomes encoding CRISPR sgRNAs are packaged into budding HIV virions, allowing robust identification of HIV restriction factors and dependency factors in a high-throughput manner. Cas9 endonuclease and sgRNA are delivered to cells in a vector that is modified to be transcribed and subsequently packaged in trans by the infecting HIV virus. Deep sequencing of packaged HIV-CRISPR RNA in nascent HIV virions released from pooled KO cells serves to proxy the efficiency of HIV replication in each genetic knockout. Thereby, our approach allows for targeted gene knockout

and a functional assay simultaneously across thousands of genes in a heterogeneous population of cells. Furthermore, as read-out of the functional assay is done of at the level of newly budded viruses, the approach allows for screening of restrictions factors affecting the full HIV life cycle. We find a small panel of ISGs to mediate IFN inhibition of HIV-1 in THP-1 cells, including MxB, TRIM5alpha, IFITM1 and Tetherin. We find that restriction factor sensitivity is, in part, strain-dependent. Further, this approach can as be used to identify HIV dependency factors as we identify CD169, SEC62 and TLR2 as host factors in THP-1 cells. The results presented here suggest that adaptation of primate lentiviruses to humans is incomplete as the same host restriction factors that block cross-species transmission also play a role in limiting the replication of highly-adapted HIV-1 in IFN-stimulated cells.

## Results

### An ISG-specific knockout screen that packages sgRNA-encoding lentiviral genomes into HIV virions

IFN$\alpha$ inhibits HIV$_{LAI}$ replication in THP-1 cells 10-fold (*Goujon et al., 2013*). To identify the factor(s) mediating the IFN$\alpha$-induced inhibition of HIV, we designed a novel HIV-based CRISPR screen in which the virus itself serves as a reporter. Cells which lack a dependency factor due to CRISPR-mediated gene knockout will release less virus, whereas cells which lack a restriction factor will produce more virus as compared to control cells which containing single-guide RNA (sgRNA) sequences that do not target any human genes, Non-Targeting Controls (NTCs). We engineered a Cas9 and sgRNA-encoding lentiviral vector such that sgRNA-encoding genomic RNA can be packaged in trans by budding HIV virions. Therefore, the normalized abundance of Cas9/sgRNA-encoding genomes themselves are the direct readout for the functional activity of each gene knockout on viral replication. Importantly, this approach allows for assay of effects of gene knockout on a complete round of viral replication.

Standard Self-Inactivating (SIN) lentiviral vectors contain a deletion in the Long-Terminal Repeat (LTR) sequence that encodes transcription factor binding sites in wild type HIV sequences. The lenti-CRISPRv2 lentiviral vector contains a SIN LTR and, therefore, full-length genomic lentiviral RNA is not transcribed after integration (*Shalem et al., 2014a*). To create a version of the lentiCRISPR vector that can be mobilized after HIV infection, we repaired the 3' LTR with a complete HIV-1 LTR. Thus, the HIV-CRISPR vector maintains complete HIV-1 LTR sequences on integration. We call this transcription- and packaging-competent construct HIV-CRISPR (*Figure 1A and B*). Importantly, in addition to internal promoters driving Cas9 and sgRNA transcription, full-length HIV-CRISPR genomic RNA is also produced after transcription from the LTR. More robust replication will occur specifically in cells that have been deleted of an antiviral factor. The enhanced release of antiviral factor-targeting HIV-CRISPR genomes occurs at a higher frequency than control sgRNAs or most sgRNAs in the library that target genes with no effect on HIV replication. Therefore, any sgRNA that targets a gene that inhibits viral replication will be enriched in the virions. Infecting HIV viruses serve to read-out efficiency of HIV replication in each knockout cell through deep sequencing of packaged HIV-CRISPR RNA in secreted HIV virions.

To target genes mediating the IFN inhibition of HIV-1, we curated a list of potential ISGs from existing microarray and RNA-seq datasets from cell types relevant to HIV-1 infection, including PBMCs, primary CD4+ T cells, monocyte-derived macrophages (MDMs), monocytes and the THP-1 monocytic cell line (*Figure 1—figure supplement 1A* and *Figure 1—source data 1*). Thus, the library is also enriched in genes that are specifically expressed in HIV target cells. For each of the 1905 ISGs present in our library, we selected a total of 8 sgRNA sequences from existing human whole-genome CRISPR/Cas9 libraries (*Figure 1—figure supplement 1B* and *Figure 1—source data 2*) (*Doench et al., 2016*; *Hart et al., 2015*; *Sanjana et al., 2014a*; *Shalem et al., 2014b*; *Shalem et al., 2015*; *Wang et al., 2015*; *Wang et al., 2014*). 200 Non-Targeting Control (NTC) sgRNA sequences that are not predicted to target any loci in the human genome were also included (*Shalem et al., 2015*) (*Figure 1—figure supplement 1B* and *Figure 1—source data 2*). In total 15,348 unique sgRNA sequences were assembled into the HIV-CRISPR backbone to create the Packageable ISG Knockout Assembly or PIKA$_{HIV}$ library (*Figure 1B*). The enrichment or depletion of sgRNA sequences in the viral RNA (vRNA) as compared to the genomic DNA (gDNA) of the cells is

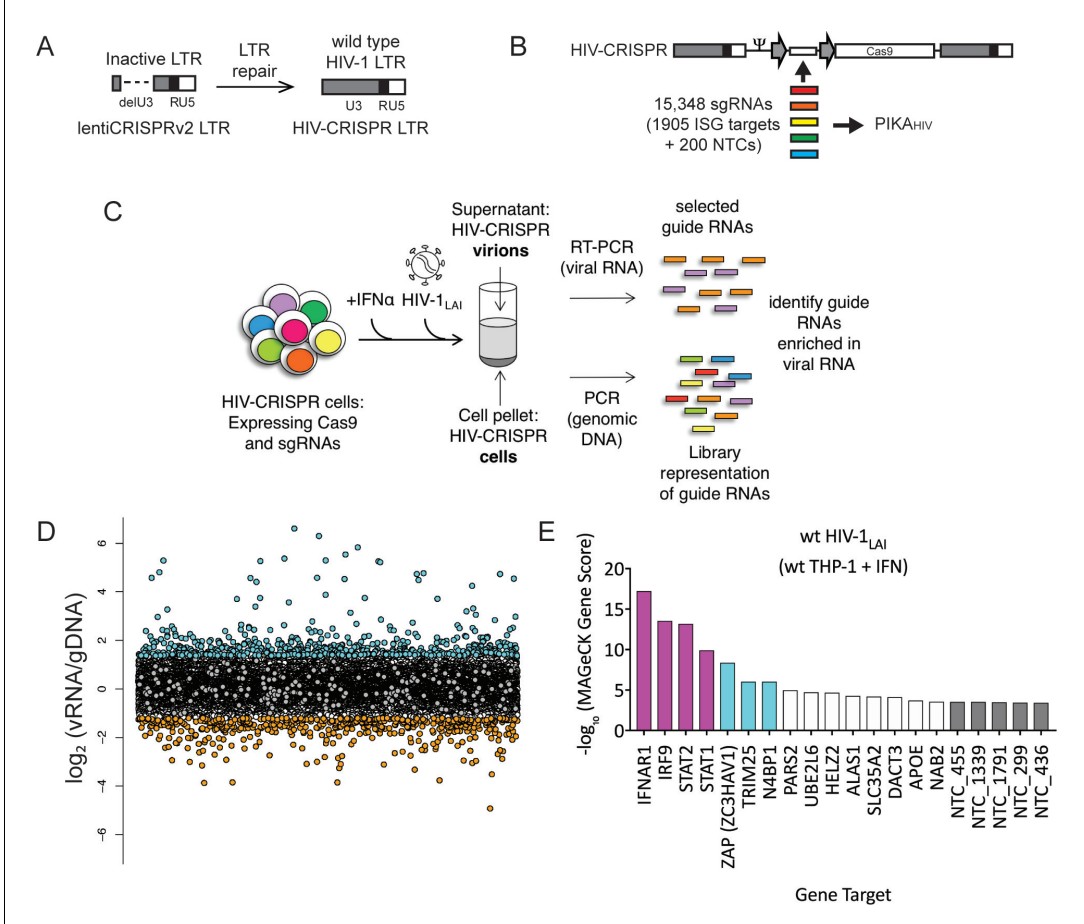

**Figure 1.** The HIV-CRISPR screen identifies gene knockouts that increase and decrease HIV infection. (**A**) The deletion in the U3 region of the SIN LTR vector (lentiCRISPRv2) was repaired by inserting the full-length HIV-1$_{LAI}$ LTR sequence to create the HIV-CRISPR construct. (**B**) The ISG-targeting sgRNA library (15,348 unique sgRNA sequences) was synthesized and assembled into the HIV-CRISPR backbone to create the PIKA$_{HIV}$ (Packageable ISG Knockout Assembly) library. (**C**) THP-1 cells containing the PIKA$_{HIV}$ CRISPR knockout library were stimulated overnight with 1000 U/mL IFNα and infected with HIV-1$_{LAI}$ in duplicate infections. Viral RNA and genomic DNA were collected 3 days post infection and sgRNA sequences present in virions (vRNA) and genomic DNA (gDNA) were quantified through RT-PCR/PCR and deep sequencing. (**D**) Average enrichment of sgRNA sequences in the viral RNA (vRNA) was compared to their representation in the sequenced genomic DNA (gDNA) (n = 2). Y-Axis: Log$_2$-normalized Fold Change (log$_2$FC) of vRNA sgRNA sequences as compared to gDNA sgRNA sequences. X-Axis: random order of individual sgRNAs. The 200 non-targeting control (NTC) sgRNAs are shown in gray. The most enriched sgRNA sequences are in cyan (top 500), most depleted in orange (bottom 500). (**E**) MAGeCK Gene analysis was performed to identify the highest-scoring genes based on sgRNA frequencies for each gene across both replicates. An NTC gene set was created in silico by iteratively binning the 200 Non-Targeting Control sgRNA sequences into NTC Genes. Y-Axis: -log$_{10}$MAGeCK Gene Score (positive). The type I IFN pathway genes IFNAR1, IRF9, STAT1 and STAT2 are shown in magenta. Non-Targeting Controls (NTCs) are in gray. Candidate Hits are cyan. The top 20-scoring genes across replicate screens are shown.

DOI: https://doi.org/10.7554/eLife.39823.003

The following source data and figure supplements are available for figure 1:

**Source data 1.** PIKA$_{HIV}$ source ISGs.
DOI: https://doi.org/10.7554/eLife.39823.006
**Source data 2.** PIKA$_{HIV}$ sgRNA sequences.
DOI: https://doi.org/10.7554/eLife.39823.007
**Source data 3.** logFC sgRNA enrichment in wt THP-1 PIKA$_{HIV}$ HIV-1$_{LAI}$ screen.
DOI: https://doi.org/10.7554/eLife.39823.008
**Source data 4.** MAGeCK.
DOI: https://doi.org/10.7554/eLife.39823.009
**Figure supplement 1.** PIKA$_{HIV}$ ISG library gene and sgRNA selection.
DOI: https://doi.org/10.7554/eLife.39823.004
**Figure supplement 2.** sgRNA read counts for the sequenced PIKAHIV Plasmid Library.

*Figure 1 continued on next page*

*Figure 1 continued*

DOI: https://doi.org/10.7554/eLife.39823.005

quantified through sequencing of sgRNA sequences both in released HIV particles and integrated into the cellular genomic DNA. sgRNAs that target antiviral genes (restriction factors) are overrepresented in viral supernatants due to more robust viral replication specifically in these KO cells. Conversely, sgRNAs that target dependency factors are depleted in viral supernatants due to decreased viral replication specifically in these KO cells.

To perform the screen, $8 \times 10^6$ THP-1 cells were transduced with the PIKA$_{HIV}$ library at an MOI <1 (MOI = 0.6) to create a population of cells with single HIV-CRISPR integrations at >500X coverage. THP-1/PIKA$_{HIV}$ cells were split in two independent replicates and left untreated or treated with IFN$\alpha$ overnight. Each replicate was then infected with HIV-1 at a dose that infects 50% of cells without IFN$\alpha$ treatment. Secreted virus was collected 3 days after infection, and sgRNA sequences encoded by HIV-CRISPR genomic RNA packaged into budding HIV virions were amplified by RT-PCR and quantitated through deep sequencing (*Figure 1C*). THP-1/PIKA$_{HIV}$ cells were collected in parallel at the time of viral supernatant harvest and the genomic DNA (gDNA) was also deep sequenced. We compared the relative enrichment of HIV-CRISPR sgRNA sequences in the viral RNA (vRNA) to the genomic DNA (gDNA) to find enriched and depleted sgRNA sequences (*Figure 1D* and *Figure 1—source data 3*). Relative to the NTCs (*Figure 1D*, gray circles), there were a number of sgRNA sequences that were either enriched (*Figures 1D* and 500 most enriched sgRNAs in cyan) or depleted (*Figures 1D* and 500 most depleted sgRNAs in orange) in the viral supernatant as compared to the NTCs. Since each gene in the PIKA$_{HIV}$ library is targeted by eight individual sgRNAs, we analyzed the enrichment across all sgRNAs for a gene using the MAGeCK package across both duplicates (*Li et al., 2014*). We identified the type I IFN pathway genes, STAT1, IFNAR1, STAT2 and IRF9 as the highest-scoring hits (magenta in *Figure 1E* and *Figure 1—source data 4*). Therefore, the PIKA$_{HIV}$ screen functioned as designed: cells in which IFN signaling is compromised exhibited increased viral production and, therefore, enriched HIV-CRISPR representation of sgRNAs in the secreted HIV virions. After the IFN pathway genes, the Zinc Antiviral Protein (ZAP) and its modifier TRIM25 were the next to highest scoring hits. ZAP is an antiviral effector that has potent activity against alphaviruses as well as moderate activity against retroviruses (*Bick et al., 2003*; *Gao et al., 2002*; *Kerns et al., 2008*; *Takata et al., 2017*). TRIM25 is a gene known to modify ZAP's antiviral activity (*Li et al., 2017*). More recently, it was shown that ZAP blocks virus replication by degrading transcripts with a high CG dinucleotide content (*Takata et al., 2017*). We also find NEDD4 Binding Protein 1 (N4BP1), a poorly-characterized inhibitor of the E3 ligase ITCH in mice (*Oberst et al., 2007*) that has not been previously known for antiretroviral activity (*Figure 1E*; cyan). N4BP1 encodes RNA binding domains and is proposed to have RNase activity (*Anantharaman and Aravind, 2006*).

## An iterative PIKA$_{HIV}$ screen in ZAP-KO cells identifies a panel of ISGs that inhibit HIV in THP-1 cells

To ask if ZAP or N4BP1 inhibit HIV replication we generated ZAP and N4BP1 knockout (KO) cell lines by electroporating crRNA/Cas9 complexes (crRNPs) into THP-1 cells followed by single-cell cloning. Knockout of either gene only very modestly increased infection by HIV-1$_{LAI}$ (*Figure 2—figure supplement 1A*; *Figure 2—figure supplement 1B*; *Figure 2—figure supplement 1C*; *Figure 2—figure supplement 1D*). Therefore, the IFN-induced restriction factors that potently inhibit HIV in THP-1 cells were not identified in this initial screen. Analysis of the CG dinucleotide content across the HIV-CRISPR genome shows high levels of CG dinucleotides, particularly in the Cas9 and Puromycin resistance ORFs, that are potential targets for ZAP-mediated RNA degradation (*Figure 2A*). Given its role in degradation of RNA with high CG content, we hypothesized that ZAP could inhibit the full-length HIV-CRISPR genomic RNA that is packaged into budding virions rather than the wt HIV genome. Thus, we determined whether or not ZAP KO allows for increased packaging of the HIV-CRISPR vector in viral particles released from cells by measuring both wild type HIV-1$_{LAI}$ genomes (HIV-1$_{LAI}$; black in *Figure 2B*) and HIV-CRISPR genomes (HIV-CRISPR; gray in *Figure 2B*) with a ddPCR assay. Indeed, we found enhanced packaging of HIV-CRISPR genomes relative to wild type

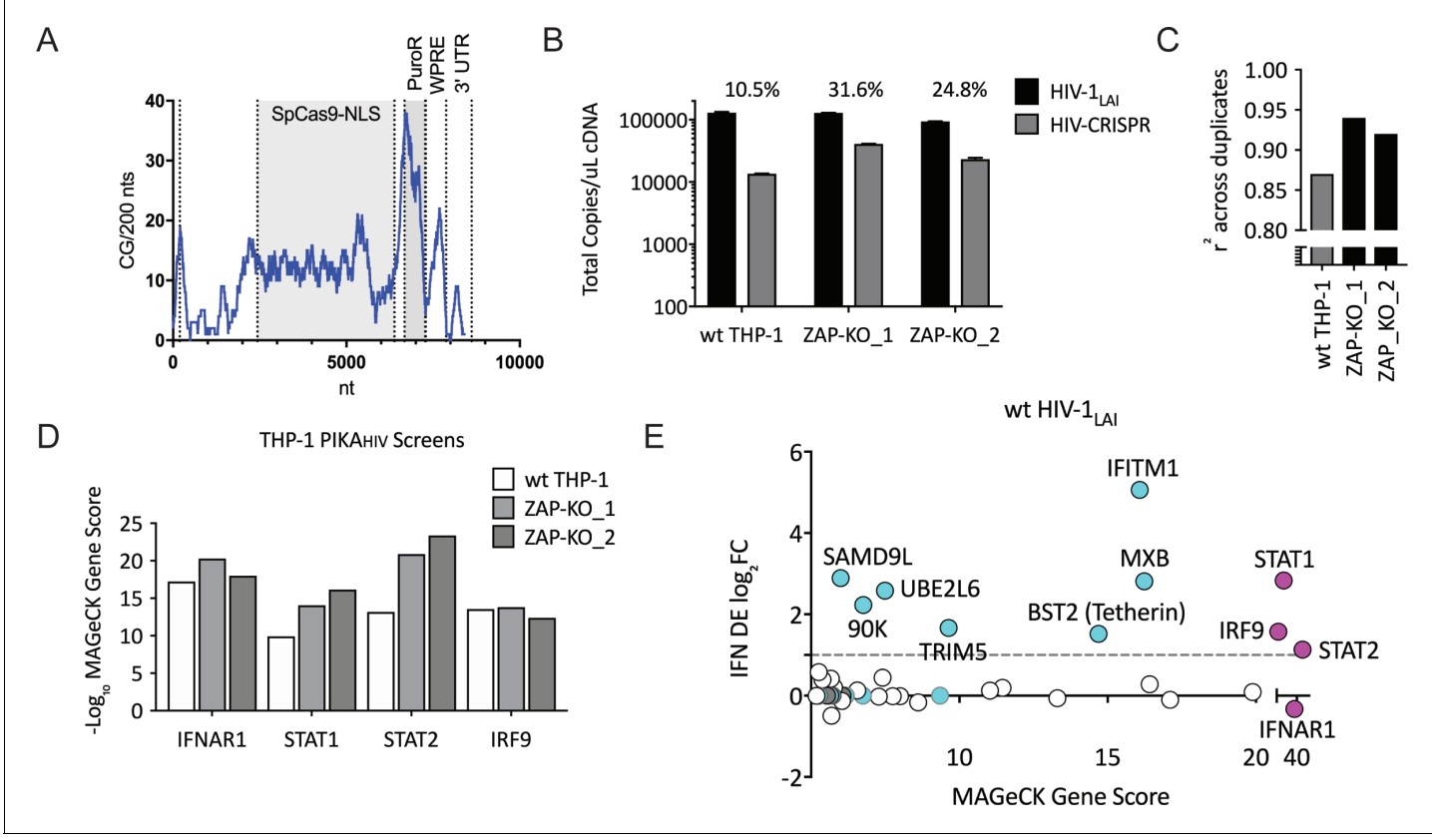

**Figure 2.** Iterative screening in ZAP-KO THP-1 cells Identifies IFNα-induced Inhibitors of HIV Replication. (A) Sliding window analysis of CG dinucleotide content per 200 nucleotides of the HIV-CRISPR construct (blue line). The Cas9 and Puromycin coding regions are shaded gray. (B) Total copies of wt HIV genomic RNA (HIV-Pol – black bars) and HIV-CRISPR genomic RNA (cPPT-U6 – gray bars) released from wild type THP-1 cells as compared to two clonal ZAP-KO THP-1 lines (ZAP-KO_1 and ZAP-KO_2) were assayed by ddPCR using primers specific for each vector template. ddPCR on cDNA from each infection was performed in duplicate. (C) PIKA$_{HIV}$ screen in IFNα-treated THP-1 cells with and without endogenous ZAP expression. Data for all sgRNAs in duplicates were compared to generate an overall r$^2$ value. Correlation of read counts across duplicate screens in each cell type (gray = wild type THP-1 cells; black = ZAP KO THP clonal lines). (D) MAGeCK Gene Scores for the type I IFN pathway genes (IFNAR1, STAT1, STAT2, IRF9) across screens in wild type THP-1 (white bars) and in two ZAP-KO clonal THP-1 lines (gray bars). (E) MAGeCK Gene Scores for Hits Identified in the THP-1 ZAP-KO cells. Positive MAGeCK Gene Scores for the results from both ZAP-KO screens was multiplied to generate a ZAP-KO MAGeCK Gene Score. Y-Axis: IFN induction (log$_2$FoldChange) in THP-1 cells calculated from GSE46599. X-Axis: Combined MAGeCK Gene Scores for top 40 Hits in both ZAP-KO screens. Magenta: IFN pathway genes (IFNAR1, STAT1, STAT2, IRF9). Cyan: highly-IFN induced, high-scoring candidate Hits. White: other high-scoring hits which were not IFN-induced (full list in *Figure 2—source data 1*). High-scoring genes with no information on IFN induction in THP-1s are plotted as IFN DE log$_2$FC = 0 but shown in Cyan with a gray outline.

DOI: https://doi.org/10.7554/eLife.39823.010

The following source data and figure supplement are available for figure 2:

**Source data 1.** THP IFN gene induction and MAGeCK Gene Analysis (Positive Scores) of ZAP-KO THP-1 PIKA$_{HIV}$ HIV-1$_{LAI}$ screens.
DOI: https://doi.org/10.7554/eLife.39823.012

**Figure supplement 1.** ZAP and N4BP1 are modest inhibitors of HIV infection.
DOI: https://doi.org/10.7554/eLife.39823.011

HIV-1$_{LAI}$ genomes in the viral supernatant in cell clones with no ZAP expression (*Figure 2B* – 10.5% in wt THP-1 cells; 24.8% and 31.6% in the ZAP-KO clonal lines).

Therefore, to circumvent the inhibitory effects of ZAP on the HIV-CRISPR vector, we repeated the PIKA$_{HIV}$ screen in two ZAP-KO THP-1 clonal cell lines (THP ZAP-KO #9* and #11* in *Figure 2—figure supplement 1A*). As expected for a screen in ZAP-KO cells, ZAP was no longer a significantly-scoring hit in the screen (rank # 1647/3812 in combined ZAP-KO screen data; *Figure 2—source data 1*). In addition, there was also no enrichment of N4BP1 or TRIM25 in the ZAP-KO screens (rank # 3789/3812 and 3090/3812 in combined ZAP-KO screen data; *Figure 2—source data 1*), suggesting that the inhibitory activity of N4BP1 and TRIM25 in the HIV-CRISPR screen are ZAP-dependent. To ask if

N4BP1 is a novel modifier of ZAP antiviral activity we screened ZAP-KO and N4BP1-KO cell lines for changes in susceptibility to Semliki Forest Virus (SFV) infection. SFV is an alphavirus that is highly-susceptible to ZAP-mediated inhibition (*Bick et al., 2003*). While ZAP KO rescued SFV replication significantly, we saw no effect of N4BP1 KO on SFV replication (*Figure 2—figure supplement 1E*). Therefore, N4BP1 may be a novel modifier of ZAP antiviral activity against lentiviruses. However, it does not appear to play a role in ZAP inhibition of alphaviruses.

Without ZAP-mediated inhibition of the HIV-CRISPR vector in ZAP-KO THP-1 cells we would expect the PIKA$_{HIV}$ screen to more efficiently identify ISGs inhibiting HIV infection. Since ZAP-KO results in increased HIV-CRISPR expression and/or packaging (*Figure 2B*) we would expect increased HIV-CRISPR representation in viral supernatants and, therefore, better sampling and therefore better correlation in read counts across duplicate infections. We assessed whether or not ZAP knockout improves performance of the HIV-CRISPR screen by analyzing read counts across duplicates in the two independent ZAP-KO THP-1 clonal lines and compared the results to the screen performed in wild-type THP-1 cells. Indeed, there was better correlation in sgRNA representation across replicates performed in ZAP-KO THP-1 cells as compared to control THP-1 cells (*Figure 2C*; $r^2 = 0.92$ and 0.94 for the ZAP-KO screens as compared to $r^2 = 0.87$ for the screen in wild type THP-1 cells). Further, an analysis specifically across the four genes that are well-described components of the type I IFN pathway also showed increased Gene Scores in the ZAP-KO THP-1 clonal lines (*Figure 2D* and *Figure 1—source data 4* and *Figure 2—source data 1*).Therefore, deletion of ZAP-mediated inhibition from THP-1 cells improves performance of the HIV-CRISPR screen.

By multiplying gene scores from both ZAP-KO screens (*Figure 2E*; MAGeCK score on x-axis) we identified a list of candidate hits. To ask which genes are most likely to contribute specifically to the IFN-mediated inhibition of HIV-1, we calculated the level of IFN induction of each of the top hits from an existing THP-1 microarray dataset (*Figure 2E*; IFN log$_2$FC on y-axis and *Figure 2—source data 1*). No hit scored as highly as the type I IFN pathway genes (magenta in *Figure 2E*). Therefore, multiple genes, rather than a single ISG, are responsible for the IFN-mediated inhibition of HIV infection in THP-1 cells. Further, a small subset of genes, including MxB, IFITM1, Tetherin, TRIM5alpha, UBE2L6, LGALS3BP/90K and SAMD9L, are candidate restriction factors mediating the IFN inhibition of HIV-1 in THP-1 cells. MxB, IFITM1, Tetherin and TRIM5alpha are the most significantly-scoring hits in the PIKA$_{HIV}$ screen that are also highly-induced by IFN (*Figure 2E*). All have well-described anti-lentiviral functions (*Goujon et al., 2013*; *Kane et al., 2013*; *Liu et al., 2013*; *Lu et al., 2011*; *Malim and Bieniasz, 2012*; *Neil et al., 2008*; *Stremlau et al., 2004*). Thus, the PIKA$_{HIV}$ screen identified IFN-induced restriction factors in a massively-parallel approach assaying all gene targets simultaneously in pools of knockout cells.

## MxB is a dominant mediator of the IFN inhibition of HIV-1 in THP-1 cells but its activity depends on the route of viral entry

To determine the relative importance of MxB to the IFN-induced block to infection, we created MxB KO THP-1 cells. MxB was deleted from THP-1 cells by transduction with a lentiCRISPRv2 MxB-targeting construct followed by single-cell cloning. Deletion of MxB expression was confirmed through western blot of IFN-treated clonal MxB-KO lines (*Figure 3A*). On creating clonal populations of THP-1 cells, we observed substantial heterogeneity across clonal lines of THP-1 cells (compare infection levels in NTC clonal lines in *Figure 3B*). Therefore, we infected many clonal NTC and MxB-KO cell lines in parallel. Infection of MxB-KO cells confirms that MxB plays a major role in the IFN block to infection as there was rescue of the IFN effect as compared to controls (*Figure 3B and C*; the Fold Inhibition in MxB-KO cells is close to 1). Therefore, MxB is a dominant, early-acting ISG inhibiting HIV replication in THP-1 cells.

The role of MxB in restriction of HIV has recently been questioned (*Opp et al., 2015*). However, in that study restriction against VSV-G pseudotyped HIV was measured rather than against a virus with a wildtype HIV envelope. Viral entry is a key target of potent IFN-mediated restriction, specifically by ISGs such as IFITMs, a family of 5 membrane-resident antiviral genes in humans with broad antiviral effects (*Shi et al., 2017*). IFITMs restrict viruses that enter cells by fusion at the plasma membrane or in the endosome. We hypothesized that sensitivity to MxB restriction may be dependent on the viral envelope since our previous work has shown that restriction of lentiviruses using distinct entry pathways are differentially affected by ISGs (*Roesch et al., 2018*). Therefore, we tested the MxB-KO cells for IFN-mediated restriction of VSV-G pseudotyped HIV-1. We found that while

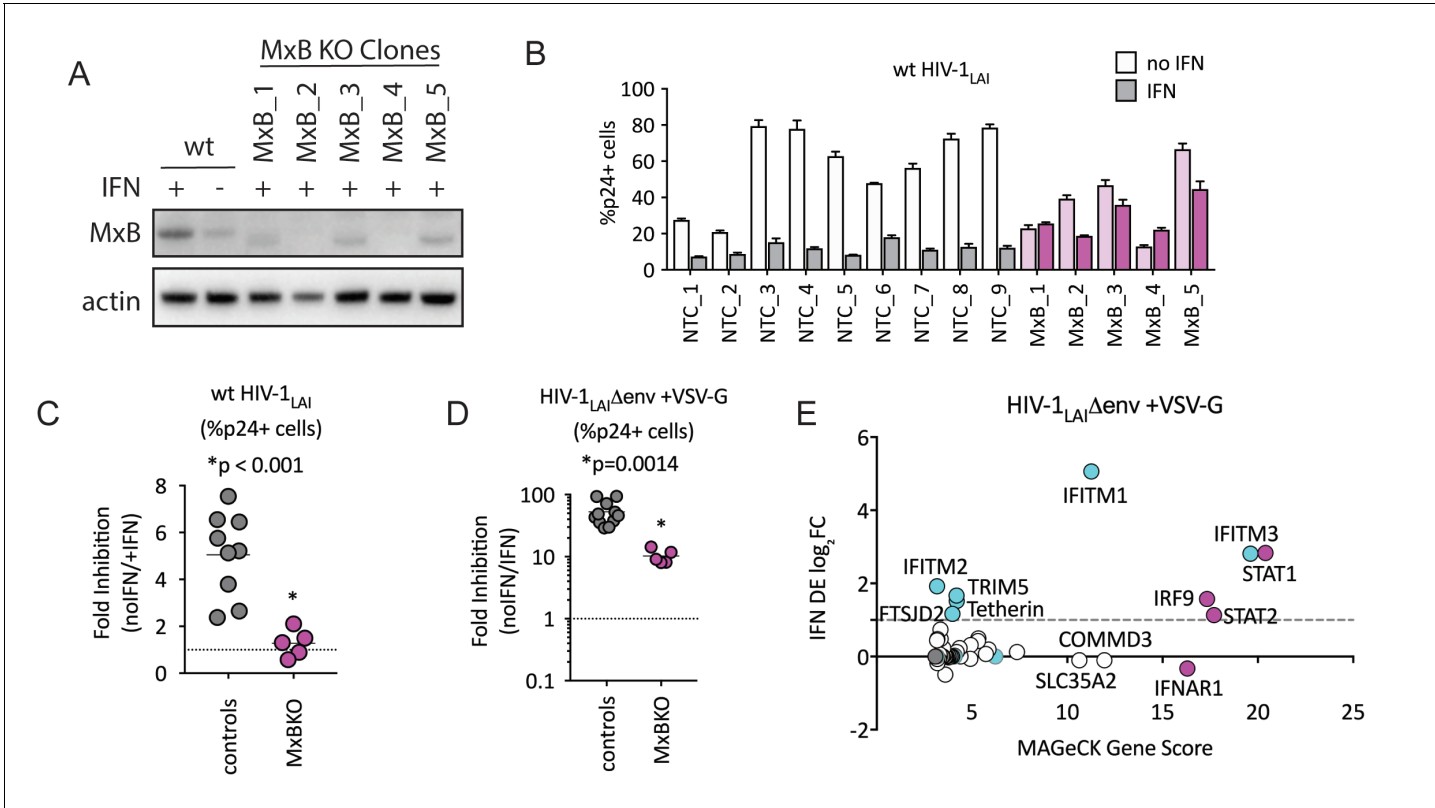

**Figure 3.** MxB is a dominant, early-acting ISG whose activity is masked by other ISGs when HIV entry is mediated by VSV-G. (**A**) Clonal MxB-KO THP-1 lines generated by transducing with an MxB-targeting sgRNA/lentiCRISPRv2 construct, selection and single-cell sorting. Western blot for MxB expression with and without IFNα stimulation overnight is shown for wild type (wt) THP-1 cells; MxB-KO clones were all IFNα treated overnight. Note: the lower molecular weight band in some lanes results from initiation at an internal Met codon that would not be predicted to have anti-HIV activity (*Goujon et al., 2015*; *Matreyek et al., 2014*). (**B**) Nine individual clonal THP-1 lines (white/gray bars) along with five clonal THP-1 MxB-KO lines (pink bars) were pre-treated with IFNα overnight and infected with wt HIV-1$_{LAI}$. The percentage of cells expressing HIV p24gag was assayed 2 days post-infection by intracellular staining and flow cytometry (n = 3). Light bars = no IFN; Dark bars = overnight IFNα treatment prior to infection. (**C**) The Fold Inhibition (%p24 +cells without IFN/%p24 +cells with IFNα) calculated for each clonal line for wt HIV-1$_{LAI}$ infections from the data in Panel B. Controls = gray; MxB-KO = magenta. Dotted line at a Fold Inhibition of 1 = no IFN inhibition. (**D**) Individual clonal control THP-1 lines (gray) along with MxB-KO clonal lines (magenta) were infected with VSV-G pseudotyped HIV both with and without IFNα pretreatment (n = 3). Fold Inhibition was calculated as in C. Dotted line: Fold Inhibition of 1 = no IFN inhibition. *p = 0.0014 (unpaired t test). (**E**) The PIKA$_{HIV}$ screen was performed in triplicate in ZAP-KO THP-1 cells. Y-Axis: IFN induction as determined by Differential Expression Analysis of microarray data in THP-1 cells (log$_2$FoldChange). X-Axis: MAGeCK Gene Scores for Top 25 Hits. Magenta: IFN pathway genes (IFNAR1, STAT1, STAT2, IRF9). Cyan: highly-IFN induced, high-scoring candidate Hits. White: non-IFN induced genes including ZAP, N4BP1, and SLC35A2. High-scoring genes with no information on IFN induction in THP-1s are plotted as IFN DE log$_2$FC = 0 but shown in Cyan with a gray outline.

DOI: https://doi.org/10.7554/eLife.39823.013

The following source data is available for figure 3:

**Source data 1.** MAGeCK Gene Analysis (Positive) of ZAP-KO THP-1 PIKA$_{HIV}$ HIV-1$_{LAI}$/VSVG Screen.

DOI: https://doi.org/10.7554/eLife.39823.014

the IFN inhibition in the MxB-KO clonal lines is significantly lower than that of control clonal lines (*Figure 3D*; p = 0.014 unpaired t test), there is still a large inhibition of replication of VSV-G pseudotyped HIV-1 by IFNα (*Figure 3D*; 53-Fold). Thus, one or more ISGs induced by IFNα potently block VSV-G mediated entry in THP-1 cells independent of MxB. To ask what factors mediate this block, we repeated the HIV-CRISPR screen with VSV-G pseudotyped HIV-1 in THP-1 cells that were knocked out for ZAP. In contrast to the screen with wildtype virus, the antiviral proteins IFITM3 is the most significantly-scoring hit, along with IFITM1, IFITM2, TRIM5alpha and Tetherin (*Figure 3E*). MxB ranks as hit #74 in this screen (*Figure 3—source data 1*) suggesting that MxB does still inhibit VSV-G pseudotyped virus but that its effect is largely masked by more potent inhibition of the VSV-G envelope by IFITM3 and other ISGs. This suggests that IFITMs are the dominant IFN-induced

blocks to replication when HIV-1 is pseudotyped with the VSV-G envelope. Significant overlap in sequence across IFITM orthologues complicates interpretation of the screen data in terms of which IFITMs are most important, as some sgRNAs in our library likely target multiple IFITM loci. However, these results show that while MxB does play a role in the IFN-mediated inhibition of VSV-G pseudo-typed HIV-1 viruses (compare MxB-KO cells to NTCs in *Figure 3D*), this effect is masked by dominant IFITM inhibition of these VSV-G pseudotyped viruses (*Figure 3E*). Similar results have been reported recently elsewhere (*Xu et al., 2018*). Therefore, viral entry route impacts restriction factor sensitivity and the role of MxB in IFN inhibition of HIV-1 in THP-1s is partially-masked by other ISGs when HIV-1$_{LAI}$ is pseudotyped by VSV-G.

## TRIM5alpha, IFITM1 and Tetherin are additional ISGs that contribute to the IFN block

We were surprised to find TRIM5alpha and Tetherin in this screen as HIV-1 is thought to be highly-adapted to these human restriction factors. To assay the contribution of each of these ISGs to IFN inhibition of HIV in THP-1 cells, we measured viral replication in THP-1 KO pools. Significant knock-out of each gene target was assayed by ICE editing analysis of amplified genomic loci (*Figure 4A*, left panel) or through cell surface staining for Tetherin (*Figure 4A*, right panel). Pretreating cells with IFNα shows ~ 7 fold inhibition of infection in the control NTC cell pools (*Figure 4B and C*) while IFN-mediated inhibition of HIV was significantly lower in MxB, TRIM5alpha and IFITM1 KO lines than in NTCs pools (*Figure 4B*; MxB_1 = 2.6 fold, MxB_2 = 2.5 fold, TRIM5_1 = 3.9 fold, TRIM5_2 = 4.8 fold, IFITM1_1 = 4.7 fold, IFITM1_2 = 6 fold and IFITM1_3 = 4.3 fold in *Figure 4C*; p < 0.05). The largest rescue we observed was in the MxB knockout pools (*Figure 4C*), confirming the key role of MxB in the IFN phenotype. However, TRIM5alpha and IFITM1 also contribute to IFNα inhibition (*Figure 4B and C*). We find no effect of Tetherin KO on early steps of HIV replication as expected given its role as a late-acting HIV restriction factor (*Figure 4B*; Tetherin_1 = 6 fold, Tetherin_2 = 6.4 fold in *Figure 4C*) (*Neil et al., 2008*; *Van Damme et al., 2008*).

We also observed a significant IFNα-mediated block to the late stages of the HIV lifecycle (after translation of the viral Gag protein used to detect infection in *Figure 4B*) in both control and MxB-KO cells (*Figure 4D*). While MxB-KO clonal lines show a decreased IFN effect compared to NTC clonal lines (compare controls to MxB-KOs in *Figure 4E*), there is still a 4.8-fold inhibition of virus released from MxB-KO clonal lines (magenta in *Figure 4E*). Since Tetherin is a well-characterized late-acting restriction factor and was also a hit in our PIKA$_{HIV}$ screen, we asked if Tetherin is responsible for the late ISG block we observed. The HIV-1 Vpu accessory gene antagonizes Tetherin-mediated restriction (*Neil et al., 2008*). Deletion of Vpu sensitizes HIV-1 to Tetherin restriction. We assayed virus release from control and Tetherin KO cell pools when IFN was added 16 hr after infection to bypass early-acting ISGs (*Figure 4F*; NTCs in gray, Tetherin KO pools in dark blue). Infection with Vpu-deficient HIV-1 (HIV-1$_{LAI}$Δvpu) in IFN-treated Tetherin-KO cells shows increased virus release as compared to control cells (Tetherin_1 = 20.5 fold, Tetherin_2 = 14 fold in *Figure 4F* – left panel), confirming the late inhibition of Vpu-deficient HIV by Tetherin. Infection of these cell pools with wt HIV-1$_{LAI}$ also shows significantly-increased virus release, suggesting that HIV-1$_{LAI}$ Vpu does not completely antagonize IFN-induced Tetherin in THP-1 cells (Tetherin_1 = 4.4 fold, Tetherin_2 = 2.14 fold in *Figure 4F* – right panel). Therefore, Tetherin is a late-acting ISG contributing to IFN inhibition of HIV-1$_{LAI}$ in THP-1 cells.

## The HIV-CRISPR screen also identifies HIV dependency factors

Although we designed our screen specifically to find IFN-induced factors restricting HIV-1 in THP-1 cells, HIV-CRISPR screening can also identify HIV dependency factors. The sgRNA sequences of genes that HIV uses for enhanced viral replication will be depleted in viral supernatants as the virus will be less well able to replicate specifically in these cells (see *Figure 1C*). Analysis of the negative MAGeCK Gene Scores, representing genes for which sgRNAs were depleted in HIV supernatants, identifies a panel of candidate host factors targeted by the PIKA$_{HIV}$ library that are important for HIV replication (*Figure 5A* and *Figure 5—source data 1*). The top hit is the HIV-1 co-receptor CXCR4 (*Figure 5A*) which is required for entry by HIV-1$_{LAI}$ (note: sgRNAs targeting the receptor, CD4, are not present in the PIKA$_{HIV}$ library).

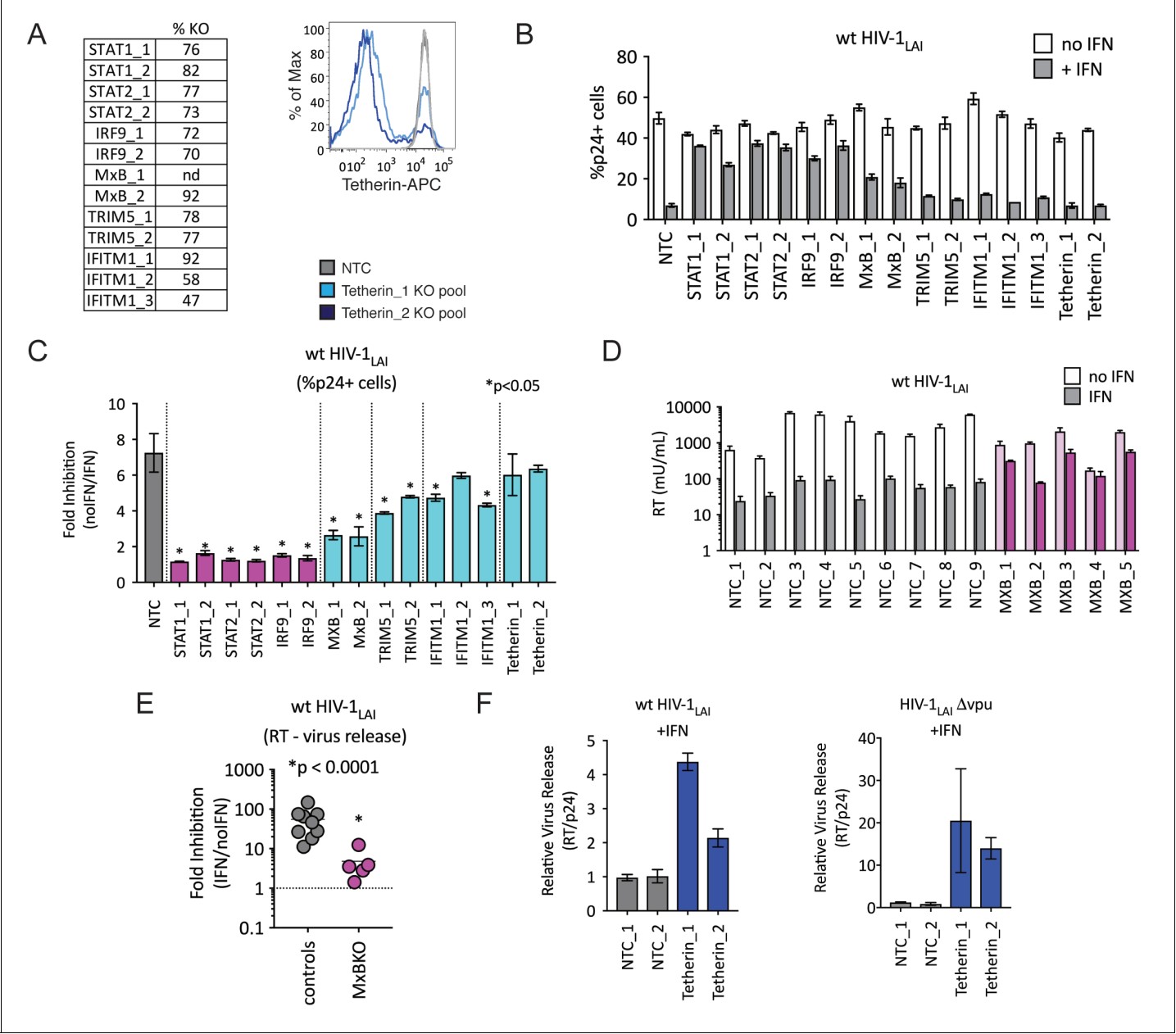

**Figure 4.** TRIM5alpha, IFITM1 and Tetherin are additional ISGs that contribute to the IFN block. (A) KO efficiencies in lentiCRISPRv2-edited THP-1 cells as determined by ICE analysis (left) or flow cytometry (right – pretreated with 1000 U/mL uIFN). (B) THP-1 cell pools edited for gene targets of interest were created by transducing wild type THP-1 cells with lentiCRISPRv2 sgRNA constructs (two different sgRNAs - for STAT1, STAT2, IRF9, MxB, TRIM5alpha and Tetherin, and three for IFITM1), selected for 2 weeks to allow gene knockout (see *Figure 4—source data 1Supplemental file 1* for analysis of gene knockout efficiency) and infected with HIV-1$_{LAI}$ with and without IFNα pretreatment in triplicate (white bars = no IFNα, gray bars = +IFNα). The percentage of cells expressing HIV p24gag was assayed 2 days post-infection by intracellular staining and flow cytometry. NTC n = 9. MxB_2 n = 6. All other pools n = 3. (C) The Fold Inhibition (%p24 +cells without IFN/%p24 +cells with IFNα) is shown for each KO pool. Control (NTC) = gray; IFN pathway genes = magenta. MxB, TRIM5alpha, IFITM1 and Tetherin = Cyan. Cells pools with significantly reduced Fold Inhibition as compared to the NTC pools *p < 0.05 (unpaired t test). (D) Virus release from the clonal NTC (white/gray) or MxB-KO clones (pink) from *Figure 3* as measured with a viral RT assay at 3 days post-infection with HIV-1$_{LAI}$ with and without IFNα (light bars = no IFN; dark bars = IFNα). (E) The Fold Inhibition (RT mU/mL without IFN/RT mU/mL with IFNα) calculated for each clonal line. Controls = gray; MxB-KO = magenta. Dotted line: Fold Inhibition of 1 = no IFN inhibition. (F) THP-1 cell pools (NTC_1 and NTC_2 = gray; Tetherin-KO pools = blue) created by transduction with lentiCRISPRv2 lentiviral vectors were infected in triplicate with Vpu-deficient HIV (HIV-1$_{LAI}$Δvpu) or wild type HIV (wt HIV-1$_{LAI}$). The HIV$_{LAI}$Δvpu contains a frameshift mutation in Vpu upstream of the env open reading frame. IFNα was added 16 hr post-infection and T-20 fusion inhibitor was added 24 hr

*Figure 4 continued on next page*

*Figure 4 continued*

post-infection. The amount of reverse transcriptase (RT) activity released into the supernatant was then normalized to the percentage of p24 +cells in order to directly quantify virus release per infected cell in the presence of IFN (RT mU/mL/%p24 +cells in culture).

DOI: https://doi.org/10.7554/eLife.39823.015

The following source data is available for figure 4:

**Source data 1.** ICE KO Editing Analysis.

DOI: https://doi.org/10.7554/eLife.39823.016

The next highest scoring hit is SIGLEC1/CD169 (*Figure 5A*), an HIV attachment factor that has been characterized to facilitate *trans* infection of CD4 +T cells by DCs through binding to sialylated glycosphingolipids on the HIV particle (*Izquierdo-Useros et al., 2012*; *Puryear et al., 2013*). CD169 is upregulated by IFNα in THP-1 cells (*Figure 5B* – gray = untreated, purple =+IFN, left panel). Our screen only assays cell-autonomous effects suggesting that CD169 also plays a role in cis-infection of monocytic cells, consistent with recent work showing enhanced infection of THP-1 cells by CD169, specifically in the presence of IFNα (*Akiyama et al., 2017*). Indeed, when CD169 expression is knocked-down (*Figure 5B*, right panel) these cells are less susceptible to infection both in the presence and absence of IFN pretreatment (*Figure 5C*), although this effect is stronger in presence of IFNα (6.5-fold vs 4.7-fold; *Figure 5C*). Thus, we find that Siglec-1/CD169 is an IFN-induced, HIV dependency factor in THP-1 cells.

TLR2, a toll-like receptor characterized to recognize bacterial PAMPs (*Akira et al., 2006*) is the next highest-scoring hit in our dependency factor analysis. We generated TLR2-KO and CXCR4-KO THP-1 cell pools by transduction with lentiCRISPRv2 sgRNA constructs and verified KO through ICE analysis of the CXCR4 genomic locus (*Figure 5D*, top) or cell surface staining of TLR2 (*Figure 5D*, bottom; NTCs in gray and TLR2-KO pools in green). Infection of these cells with wildtype HIV-1$_{LAI}$ demonstrates lower infection as compared to the controls, although this effect is not as extreme as for CXCR4 KO (*Figure 5E* left panel; 31-fold decreased infection for CXCR4-KO lines in orange, 3-fold for TLR2-KO lines in green). Of note, infection with VSV-G pseudotyped HIV-1 shows a loss of infectivity similar to wild-type HIV-1 (*Figure 5E* bottom right panel). Therefore the effect of TLR2 on enhanced infection is independent of viral entry. Conversely, the CXCR4-KO lines are efficiently infected by VSV-G pseudotyped HIV-1 (orange in *Figure 5E*, right panel) as entry occurs independent of the CXCR4 co-receptor.

Finally, we investigated the effect of SEC62, on HIV replication. SEC62 is a component of the protein translocation machinery in the ER membrane. Knockdown of SEC62 by transducing THP-1 cells with two lentiviral shRNA constructs targeting SEC62 showed significant loss of expression as measured by Western Blot (*Figure 5F*). Infection of these cells with wildtype HIV-1$_{LAI}$ showed decreased levels of infection (*Figure 5G*, left panel). Therefore, SEC62 is a dependency factor for HIV replication in THP-1 cells. As SEC62 is a component of the machinery that mediates translocation of transmembrane proteins into the ER membrane for targeting to the cell surface, we reasoned that SEC62 knockdown may be affecting cell-surface expression of HIV receptors, co-receptors or other cell-surface markers mediating attachment and/or entry of HIV. Consistent with this hypothesis, infection via an alternative entry pathway via pseudotyping HIV-1 particles with VSV-G, demonstrates equivalent infection in control and SEC62 knockdown cells (*Figure 5G*, right panel). Analysis of the cell surface expression of the HIV-1 receptor, CD4, shows that levels of CD4 on the cell surface are decreased in SEC62 knockdown cells (*Figure 5H*, left panel). Interestingly, we do not observe decreased cell-surface expression of CXCR4 (*Figure 5H*, right panel), suggesting that the effect of SEC62 knockdown on cell surface proteins is not global but specific to certain transmembrane proteins.

## PIKA$_{HIV}$ screen with an R5-tropic HIV-1 subtype A virus

To ask if the same restriction factor and dependency factor genes are important for replication of a primary HIV isolate, we repeated the PIKA$_{HIV}$ screening with a CCR5-tropic Clade A HIV-1 virus derived from a molecular clone containing sequences of two primary HIV isolate,HIV-1$_{Q23}$ and HIV-1$_{BG505}$, here called HIV-1$_{Q23.BG505}$ (*Haddox et al., 2018*; *Poss and Overbaugh, 1999*). In order to do this screen, we engineered the THP-1 ZAP KO cells to also express the HIV co-receptor CCR5 (see Materials and methods). Similar to our initial screening with HIV-1$_{LAI}$ the type I IFN pathway

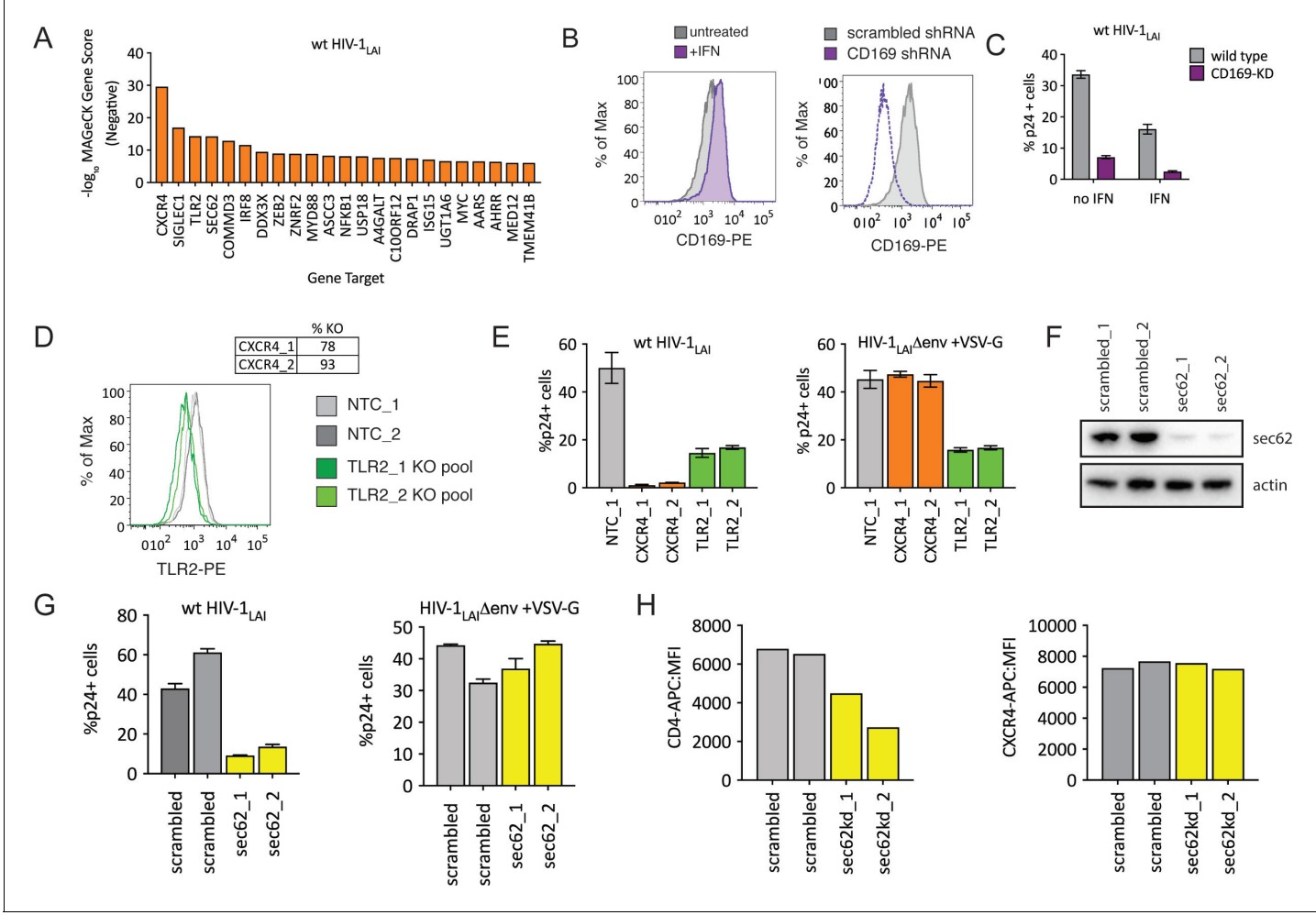

**Figure 5.** HIV-CRISPR Screening Identifies HIV Dependency Factors. (**A**) Negative MAGeCK Gene Scores across both ZAP-KO Screens ranked from most depleted genes on the X-axis. Only the top 25 hits are shown. (**B**) Left: THP-1 cells were stimulated overnight with IFNα and assayed for cell surface SIGLEC1/CD169 expression by flow cytometry. Right: Control (scrambled - gray) THP-1 cells and THP-1 cells transduced with a SIGLEC1/CD169-targeting shRNA construct (dotted purple line) were assayed for cell surface SIGLEC1/CD169 expression after overnight IFNα treatment. (**C**) Infection of control (gray – wild type) and SIGLEC1/CD169 knockdown THP-1 (purple - CD169-KD) with and without IFNα (1000 U/mL u IFNα) and assayed by intracellular p24gag 2 days after infection (**D**) KO efficiency as determined by ICE analysis (CXCR4) or flow cytometry (TLR2). (**E**) Infection of control (gray – NTC), CXCR4-KO pools (orange) and TLR2-KO pools (green) were assayed for the % of cells expressing HIV p24gag 2 days post-infection by intracellular staining and flow cytometry. Left: wt HIV-1$_{LAI}$ (n = 3). Right: HIV-1$_{LAI}$Δenv + VSV G (n = 3). (**F**) SEC62 knockdown after transduction with two LKO SEC62 shRNA constructs. Western blot of the sec62-targeting shRNA cell lines is shown together with two control (scrambled in gray) cell lines. Loading control = actin. (**G**) Infection of SEC62-KD (yellow) and control (scrambled in gray) with wt HIV-1$_{LAI}$ (left panel) or HIV-1$_{LAI}$Δenv + VSV G (right panel). The % of cells expressing HIV p24gag 2 days post-infection is shown. (**H**) The mean fluorescence intensity (MFI) of CD4-APC (left panel) and CXCR4-APC (right panel) cell surface staining of control (scrambled in gray) and SEC62-KD (yellow) THP-1 cell pools.

DOI: https://doi.org/10.7554/eLife.39823.017

The following source data is available for figure 5:

**Source data 1.** MAGeCK Gene Analysis (Negative Scores) of ZAP-KO THP-1 PIKA$_{HIV}$ HIV-1$_{LAI}$ screens.

DOI: https://doi.org/10.7554/eLife.39823.018

genes are the most significantly-scoring genes for HIV-1$_{Q23.BG505}$ (**Figure 6A** and **Figure 6—source data 1**). We also find MxB, TRIM5alpha, IFITM1 and UBE2L6 to be among the top hits in this screen in common with HIV-1$_{LAI}$ (**Figure 6A** and **Figure 6—source data 1**). To directly visualize the similarities and differences for restriction factor hits from the screens between the two different HIV strains, we replotted the data of the MaGeCK scores for the top 30 scoring genes for HIV-1$_{LAI}$ on the X axis and for the top 30 scoring genes for HIV-1$_{Q23.BG505}$ on the Y axis (**Figure 6B**). We find IFI16, a

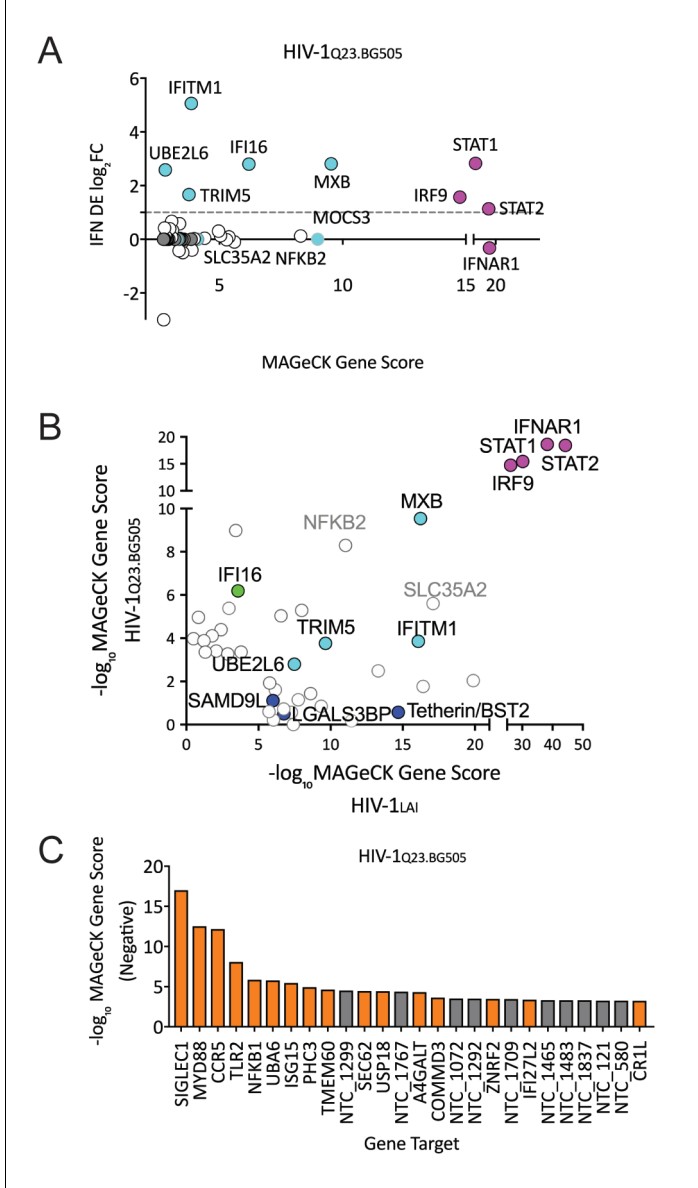

**Figure 6.** HIV-CRISPR Screen of an R5-tropic Clade 1 Isolate (HIV-1$_{Q23.BG505}$). (**A**) The PIKA$_{HIV}$ screen was performed in duplicate in ZAP-KO THP-1 cells with HIV-1$_{Q23.BG505}$. Y-Axis: IFN induction as determined by Differential Expression (DE) Analysis of microarray data in THP-1 cells (IFN DE log$_2$FC). X-Axis: MAGeCK Gene Scores for Top 50 Hits. Magenta: IFN pathway genes (IFNAR1, STAT1, STAT2, IRF9). Cyan: highly-IFN induced, high-scoring candidate Hits. White: non-IFN induced genes. High-scoring genes with no information on IFN induction in THP-1s are plotted as IFN DE log$_2$FC = 0 but shown in Cyan with a gray outline. (**B**) Comparison of the top 30 gene hits from either the HIV-1$_{LAI}$ screen (X-axis) or the HIV-1$_{Q23.BG505}$ screen (Y-axis). Magenta: IFN pathway genes. Cyan: highly-IFN induced, high-scoring genes for both viruses. Green: highly-IFN induced and high-scoring for HIV-1$_{Q23.BG505}$. Dark Blue: highly-IFN induced and high-scoring for HIV-1$_{LAI}$. Gray outlines are genes that are not significantly upregulated by IFN. (**C**) Negative MAGeCK Gene Scores for the HIV-1$_{Q23.BG505}$ PIKA$_{HIV}$ screen. Only the top 25 hits are shown. Gray = NTCs; Orange = previously described or novel candidate HIV dependency factors.

DOI: https://doi.org/10.7554/eLife.39823.019

The following source data and figure supplement are available for figure 6:

**Source data 1.** MAGeCK Gene Analysis (Negative) of ZAP-KO THP-1 PIKA$_{HIV}$ HIV-1$_{LAI}$/VSVG Screen.
DOI: https://doi.org/10.7554/eLife.39823.021

**Figure supplement 1.** HIV Dependency Factors for VSV-G pseudotyped HIV-1.
DOI: https://doi.org/10.7554/eLife.39823.020

cytoplasmic DNA sensor known to inhibit HIV (*Jakobsen et al., 2013*; *Monroe et al., 2014*), as a restriction factor specific to HIV-1$_{Q23.BG505}$ (*Figure 6B* and *Figure 6—source data 1*). In contrast, Tetherin is no longer a significantly scoring hit in the HIV-1$_{Q23.BG505}$ screen (rank #1897/3812; *Figure 6—source data 1*), suggesting that Tetherin plays less of a role in restriction of this primary isolate as compared to HIV-1$_{LAI}$. Finally, SLC35A2 and NFKB2, while not IFN-induced in THP-1s, are highly-scoring genes for both viruses (*Figure 6B*). SLC35A2, a nucleotide sugar transporter (reviewed in *Hadley et al., 2014*), and NFKB2, a component of the NFKB transcription factor complex (reviewed in *Vallabhapurapu and Karin, 2009*), may both negatively impact HIV replication in an IFN-independent manner.

Next, we examined the HIV-1$_{Q23.BG505}$ data to identify HIV dependency factors important for infection of this virus (*Figure 6C* and *Figure 6—source data 1*). Similar to our HIV-1$_{LAI}$ screen we find the coreceptor (in this case CCR5) to be one of the most significantly-scoring depleted genes in the HIV-1$_{Q23.BG505}$ screen (*Figure 6C* and *Figure 6—source data 1*). SIGLEC1/CD169 is the highest hit in the HIV-1$_{Q23.BG505}$ screen (*Figure 6C* and *Figure 6—source data 1*) suggesting that this HIV attachment factor is important for infection also by a Clade A primary isolate. In common with HIV-1$_{LAI}$ we also find SEC62, TLR2, MYD88 and NFKB1 again as important HIV dependency factors for HIV-1$_{Q23.BG505}$ (*Figure 6B* and *Figure 6—source data 1*). In addition, we analyzed the VSV-G pseudotyped HIV-1$_{LAI}$ infection to compare requirements across viral entry routes (*Figure 6—figure supplement 1*). The role of SEC62 in CD4 receptor expression is further supported by our finding of SEC62 as a hit for both HIV envelope-containing viruses but not for the VSV-G pseudotyped infection (compare *Figures 5A* and *6C* and *Figure 6—figure supplement 1*).

## Discussion

We have designed and validated a novel CRISPR knockout screening approach in which the virus itself serves to report levels of infection to identify genes important for HIV infection. Subsequent deep sequencing can quantitate effects of individual gene knockouts on viral replication in a massively-parallel fashion. Using the PIKA$_{HIV}$ library targeting human ISGs, we demonstrate that the IFN-mediated inhibition of HIV in THP-1 cells is due to the combined action of a small panel of ISGs that includes known HIV restriction factors like MxB, TRIM5alpha, IFITM1 and Tetherin. Each of these ISGs individually imposes only modest restriction of HIV replication but together mediate robust restriction of HIV replication.

### Incomplete antagonism of HIV restriction factors

While HIV has evolved antagonism or evasion strategies for restriction factors that limit replication in cells from other species, the results here imply that HIV is not able to completely antagonize or escape some host encoded restriction factors. This is true even for well-adapted HIV strains, such as HIV$_{LAI}$, that were isolated from chronically-infected individuals with high viral load. Such incomplete antagonism may be due to conflicting evolutionary pressures acting on the HIV genome. HIV proteins must evade restriction factor recognition while maintaining other protein functions, including evasion of other restriction factors and binding to required cellular host factors.

Our finding of significant TRIM5alpha restriction in human cells suggests that HIV is still partially-sensitive to human TRIM5alpha-mediated restriction. Similarly, we find that IFN-induced MxB restricts infection in THP-1 cells, consistent with previous work (*Goujon et al., 2013*; *Kane et al., 2013*; *Liu et al., 2013*). A similar example of incomplete antagonism of human restriction factors can be found in HIV-infected patients in which a signature of APOBEC3 G-to-A hypermutation in integrated proviruses can be observed (*Cuevas et al., 2015*; *Sadler et al., 2010*) despite the fact that HIV encodes an antagonist, Vif, that targets APOBEC3 proteins for degradation.

Both TRIM5alpha and Tetherin are rapidly-evolving genes in primates with described consequences for host adaptation by primate lentiviruses (*Lim et al., 2010*; *Liu et al., 2005*). Capsids from diverse primate lentiviruses have adapted to TRIM5alpha alleles in various primates and are variably-restricted by TRIM5alpha orthologs (*Kirmaier et al., 2010*). Selection for capsid mutations that evade TRIM5alpha restriction is a key adaptive step that HIV and related SIVs must make to successfully replicate in a particular primate species (*Wu et al., 2013*). Consistent with a role for TRIM5alpha in humans, TRIM5alpha is active against HIV in Langerhans cells (*Ribeiro et al., 2016*) and TRIM5alpha polymorphisms are associated with the clinical course of HIV-1 infection (*van Manen et al.,*

*2008*). Further, CA mutations in HIV-infected individuals have been associated with sensitivity to TRIM5alpha restriction (*Battivelli et al., 2010*; *Onyango et al., 2010*). Also of note, most studies of HIV have not been done in the context of IFN despite evidence that TRIM5alpha is highly-IFN upregulated in HIV target cells (*Carthagena et al., 2009*). Thus, our finding of IFN-mediated TRIM5alpha inhibition of HIV represents a potentially-important role of TRIM5alpha particularly during acute infection when IFN levels are high. More stable artificial variants of human TRIM5alpha can inhibit HIV-1 (*Richardson et al., 2014*), suggesting that increased TRIM5alpha levels, such as after IFN induction, may play a role in restricting HIV replication. Evasion of TRIM5alpha restriction may come at the cost of loss of fitness due to other requirements for capsid function within host cells such as uncoating, nuclear import and integration. Further, capsid sequences also mediate evasion of other restriction factors, including MxB, or escape from host CTL responses. Primary isolates of HIV-1 have increased sensitivity to TRIM5alpha that is proposed to be driven by CTL escape variants (*Battivelli et al., 2011*). Therefore, we speculate that this TRIM5alpha sensitivity may underscore the requirement of HIV proteins to balance multiple functions simultaneously to infect human cells. Like TRIM5alpha, escape from IFITMs also appears to be subject to conflicting evolutionary pressures. IFITMs may exert significant selective pressure in vivo as HIV evolves increased susceptibility to IFITMs over the course of infection (*Foster et al., 2016*). In addition to basic required functions of HIV proteins, we further propose that further adaptation of HIV for efficient replication in human cells may not be possible due to constraints on viral evolution imposed simultaneously by multiple, independent restriction factor barriers.

## Comparison of restriction factors against different viral strains using HIV-CRISPR screens

MxB, TRIM5alpha, UBE2L6 and IFITM1 are strong hits in both the HIV-1$_{LAI}$ and the HIV-1$_{Q23.BG505}$ screens (cyan in *Figure 6B*) suggesting that these ISGs inhibit a range of HIV isolates. UBE2L6 has recently been shown to inhibit HIV in an overexpression assay and is antagonized by Vpu (*Jain et al., 2018*). In contrast, several other ISGs scored highly in the HIV-1$_{LAI}$ PIKA$_{HIV}$ screen but not the HIV-1$_{Q23.BG505}$ screen including LGALS3BP (also known as 90K or M2BP), SAMD9L and Tetherin/BST2 (dark blue in *Figure 6B*). UBE2L6 and 90K inhibit HIV in over-expression assays (*Jain et al., 2018*; *Lodermeyer et al., 2013*; *Wang et al., 2016*) and SAMD9L was recently shown to be an IFN-induced restriction factor for poxviruses (*Meng et al., 2018*). Tetherin/BST2, LGALS3BP/90K/M2BP and SAMD9L therefore show differential, strain-dependent restriction. Finally, the HIV-1$_{Q23.BG505}$ screen reveals one factor, IFI16, that is a strong hit in the HIV-1$_{Q23.BG505}$ screen but not the HIV-1$_{LAI}$ screen (Green in *Figure 6B*). IFI16 is a dsDNA sensor that has been implicated in sensing and inhibition of HIV infection (*Jakobsen et al., 2013*; *Monroe et al., 2014*). Our results here suggest that IFI16 restricts HIV-1 in THP-1 cells in a strain-dependent manner.

Similar to TRIM5alpha, our finding of inhibition of wt HIV by Tetherin despite intact Vpu expression may suggest a functional tradeoff in which HIV Vpu is unable to completely antagonize host cell Tetherin activity. It has previously been shown that primary HIV isolates better antagonize Tetherin as compared to HIV-1 NL4-3 (*Pickering et al., 2014*) which may explain why Tetherin is a hit in the screen using HIV-1$_{LAI}$, but not HIV-1 $_{Q23.BG505}$. Adaptation HIV-1 group M to the unique form of human Tetherin allele required evolution of the viral protein Vpu to antagonize Tetherin (*Kmiec et al., 2016*; *Lim et al., 2010*; *Sauter et al., 2009*). Consistent with conflicting evolutionary constraints, IFN treatment in HCV- and HIV- coinfected patients resulted in evolution of Vpu variants with stronger Tetherin antagonism when ISGs are expressed in vivo (*Pillai et al., 2012*). Perhaps more complete antagonism of Tetherin by Vpu would compromise some of the other functions of Vpu in cells (*Apps et al., 2016*; *Margottin et al., 1998*; *Schubert et al., 1998*; *Shah et al., 2010*). Further, a moderate level of Tetherin antagonism could be selected for if cell-to-cell transmission is enhanced by Tetherin restriction (*Gummuluru et al., 2000*; *Jolly et al., 2010*), such as is observed for MoMLV in mice (*Liberatore et al., 2017*). Interestingly, Vpu counteracts UBE2L6 (*Jain et al., 2018*) which we describe as an inhibitor of both HIV-1$_{LAI}$ and HIV-1$_{Q23.BG505}$ suggesting that Vpu from both viruses are unable to fully counteract UBE2L6 in IFN-treated THP-1 cells.

## ZAP inhibition of a lentiviral vector

We found that ZAP mediates a small, but detectable inhibition of HIV replication as we find enhanced infection of ZAP-KO cells both in the presence and absence of IFN pretreatment (*Figure 2 - Supplemental Figure 1A and 1B*). Similar to the effect of ZAP, N4BP1 (Nedd4-binding protein 1) also has a modest effect on HIV replication both after IFN pretreatment and when constitutively-expressed (*Figure 2 - Supplemental Figure 1C and 1D*). In our screen, the anti-lentiviral function of N4BP1 appears to be genetically linked to ZAP activity as N4BP1 is no longer a hit in the ZAP-KO screen (*Figure 2 – Supplemental Figure 1E*). Therefore, N4BP1 may modify or enhance ZAP-mediated antiviral activity similar to the modification of ZAP activity described for TRIM25 (*Li et al., 2017*). However, we did not find a role for N4BP1 in ZAP-mediated inhibition of SFV (see *Figure 2 – Supplemental Figure 1E*). Therefore, the role of N4BP1 in ZAP's antiviral activity varies depending on the viral target.

## The HIV-CRISPR screening approach

Other high-throughput screens with siRNA pools have focused on identifying dependency factors that the virus takes advantage of to infect cells (*Brass et al., 2008*; *König et al., 2008*; *Zhou et al., 2008*). Importantly, all of these high-throughput approaches still require individual gene knockdowns or overexpression in individual wells. The HIV-CRISPR screening approach represents a significant advance in screening for host factors that affect HIV replication in several ways, including: (1) we can simultaneously screen thousands of gene targets in a single experiment, (2) we can use any virus strain, (3) we do not need any type of reporter to assay infections as virus replication itself provides the assay readout and (4) we can capture host factors that affect all stages of the HIV life-cycle including entry, nuclear import, integration, transcription, nuclear export, translation, packaging, budding and release. After finding ZAP-mediated inhibition of the HIV-CRISPR vector used in our screening approach, we modified our PIKA$_{HIV}$ screen to avoid this inhibition by specific KO of ZAP expression and rescreening in ZAP-KO THP-1 clonal lines. Genetic deletion of ZAP resulted in enhanced performance of the HIV-CRISPR screen and allowed for our identification of ISGs contributing to the IFN block in THP-1 cells. Further the data presented here demonstrates that the screen is sensitive enough to find key factors in just a single round of viral replication, even when multiple factors together mediate potent inhibition.

## HIV-CRISPR screening can identify HIV dependency factors

Despite targeting less than 10% of the genes in the human genome by our PIKA$_{HIV}$ library, we were able to identify and validate a small panel of HIV dependency factors that HIV usurps to enhance infection in THP-1 cells. We demonstrate that the HIV attachment factor, SIGLEC1/CD169, plays a role in enhancing infection in THP-1 cells in cis rather than the more fully described role of SIGLEC1 to mediate infection from dendritic cells to T cells in trans (*Izquierdo-Useros et al., 2012*; *Puryear et al., 2013*). Further, we find that TLR2 mediates enhanced infection of THP-1 cells by HIV-1 regardless of viral entry pathway used, as it impacted infection through both the HIV envelope and the VSV-G glycoprotein (*Figure 5E*). Recent work in CD4 +T cells has similarly demonstrated enhanced infection and/or viral production in T cells on stimulation of TLR2 (*Bolduc et al., 2017*; *Ding and Chang, 2012*; *Ding et al., 2010*; *Equils et al., 2003*; *Henrick et al., 2015*). Of note, MYD88, a downstream effector for TLR2 activation of transcription, is also a strong hit in our dependency factor screening (*Figure 5A*) suggesting that it is the downstream signaling functions of TLR2 that enhance infection. In addition to CD169 and TLR2, our identification of SEC62 as a novel HIV dependency factor that correlates with CD4 receptor cell surface expression highlights the ability of the HIV-CRISPR screening approach to find genes that function in pathways (such as CD4 receptor expression) important for HIV infection. Finally, as would be expected, NFKB1 is a dependency factor (*Figure 5A*) as it is a precursor to a downstream activator of the HIV LTR that is important for HIV infection, although somewhat surprisingly NFKB2 is a restriction factor in the screen (*Figure 6B*). Thus, HIV-CRISPR screens have the capability of identifying pathways of both positive and negative HIV regulation.

## Comparison with other HIV CRISPR screens

A recent CRISPR knockout screen demonstrated that a pooled CRISPR approach to gene knockout could identify HIV dependency factors (*Park et al., 2017*). The Park *et al.* study differs from our study in several key aspects. The screen by Park *et al.* relied on Tat-driven LTR-GFP reporter gene expression as well as many rounds of spreading infection across multiple weeks in culture in a T cell line. In contrast, PIKA$_{HIV}$ screening is performed over a single round of infection in three days in the THP-1 monocytic cell line with the screen relying on virus replication itself to enrich for gene targets of interest rather than selecting on a reporter gene. Further, Park et al screened with a whole-genome library while our screening has been performed with an ISG-specific sgRNA library (PIKA$_{HIV}$). Moreover, we screened in the presence of IFN with a specific focus on finding HIV restriction factors, whereas Park et al. specifically screened for HIV dependency factors. Both screens do identify the appropriate HIV co-receptor (CXCR4 or CCR5 in our study and CCR5 in the Park et al. study). Of note, 3 of the five genes identified by Park et al. (TPST2, SLC35B2 and CD4) are not represented in the PIKA$_{HIV}$ library and, therefore, could not be identified in our screen. The final gene identified by Park et al., ALCAM, is not found as a hit in our screening and may reflect important differences in cell type or other technical aspects of the screening approaches. For example, the genes identified by Park et al. are the HIV receptor or co-receptor (CD4, CCR5), genes that affect CCR5 expression (TPST2 and SLC35B2) or genes that affect the ability of cells to complex with other cells (ALCAM). It is possible that the Park et al. screen may, therefore, reflect the loss of the ability of these KO cells to support syncytia formation rather than a loss of HIV infection per se. Some hits identified in our PIKA$_{HIV}$ screening were not found in the Park et al. study, including SEC62, TLR2 and SIGLEC1/CD169. We hypothesize that differences in cell type, viral strains and screening approach likely explain these differences. Further studies using a whole genome HIV-CRISPR library should more comprehensively identify further HIV dependency factors that are not present in the PIKA$_{HIV}$ library.

In summary, here we describe a novel screen that is highly sensitive to detect restriction factors for HIV-1. This new tool shows that the IFN inhibition of HIV-1 in a monocytic cell line is due the combined function of fewer than eight different genes. Our results demonstrate that IFN-mediated inhibition of HIV-1 in THP-1 cells is mediated by restriction factors for which HIV has described mechanisms of antagonism and/or escape. The increased IFN sensitivity of specific HIV strains, such as those isolated during chronic HIV infection, may be due to relaxation of constraints on the virus that would otherwise limit virus replication during transmission events. We propose that conflicting functional constraints acting on HIV may result in incomplete antagonism or escape from host ISGs during chronic infection.

# Materials and methods

**Key resources table**

| Reagent type (species) or resource | Designation | Source or reference | Identifiers | Additional information |
|---|---|---|---|---|
| Cell line (homo sapiens) | THP-1 | NIH AIDS Reagent Program | 9942; RRID: CVCL_0006 | |
| Cell line (homo sapiens) | THP-1 ZAP-KO | this paper | | Progenitor: THP-1 |
| Cell line (homo sapiens) | THP-1 N4BP1-KO | this paper | | Progenitor: THP-1 |
| Cell line (homo sapiens) | THP-1 MxB-KO | this paper | | Progenitor: THP-1 |
| Cell line (homo sapiens) | 293T | ATCC | CRL-3216; RRID: CVCL_0063 | |
| Cell line (homo sapiens) | TZM-bl | NIH AIDS Reagent Program | 8129; RRID: CVCL_B478 | |
| Antibody | MxB (goat polyclonal) | Santa Cruz Biotechnologies | sc-271527; RRID: AB_10649506 | 1:200 |

*Continued on next page*

*Continued*

| Reagent type (species) or resource | Designation | Source or reference | Identifiers | Additional information |
|---|---|---|---|---|
| Antibody | ZAP/ZC3HAV1 (rabbit polyclonal) | Proteintech | 16820–1-AP; RRID: AB_2728733 | 1:5000 |
| Antibody | N4BP1 (rabbit polyclonal) | Cohesion Biosciences | CPA2415 | 1:1000 |
| Antibody | SEC62 (rabbit polyclonal) | Abcam | ab168843 | 1:2000 |
| Antibody | actin (rabbit polyclonal) | Sigma | A2066; RRID: AB_476693 | 1:5000 |
| Antibody | tubulin (mouse monoclonal) | Sigma | T6199; RRID: AB_477583 | 1:1000 |
| Antibody | goat anti-rabbit IgG-HRP | Santa Cruz Biotechnologies | sc-2004; RRID: AB_631746 | 1:5000 |
| Antibody | donkey anti-goat HRP | Santa Cruz Biotechnologies | SC-2020; RRID: AB_631728 | 1:5000 |
| Antibody | CD169 (Siglec-1) (mouse monoclonal) | NOVUS | NB600-534; RRID: AB_2189038 | 1:50 |
| Antibody | APC Mouse anti-Human CD4 | BD Pharmingen | 555349; RRID: AB_398593 | 1:50 |
| Antibody | CXCR4 | eBioscience | 17-9999-42; RRID: AB_1724113 | 1:50 |
| Antibody | PE anti-human CD282 (TLR2) | BioLegend | 309707; RRID: AB_314777 | 1:100 |
| Antibody | APC anti-human CD317 (BST2/Tetherin) | BioLegend | 348410; RRID: AB_2067121 | 1:50 |
| Antibody | KC57-FITC | Beckman Coulter | 6604665; RRID: AB_1575987 | |
| Sequence-based reagent | ZAP crRNA (ATGTGGAGT CTTGAACACGG) | IDT | this paper | |
| Sequence-based reagent | tracrRNA | IDT | 1072534 | |
| Recombinant DNA reagent | lentiCRISPRv2 (plasmid) | Addgene | 52961 | |
| Recombinant DNA reagent | HIV-CRISPR (plasmid) | this paper | | Progenitors: lentiCRISPRv2; Genscript synthesis |
| Recombinant DNA reagent | PIKA-HIV (plasmid library) | this paper | | Progenitors: PCR (synthesized oligos); HIV-CRISPR |
| Recombinant DNA reagent | pMD2.G | Addgene | 12259 | |
| Recombinant DNA reagent | psPAX2 | Addgene | 12260 | |
| Recombinant DNA reagent | LKO Sec62 shRNA | Sigma | TRCN000 0289739 | |
| Recombinant DNA reagent | LKO Sec62 shRNA | Sigma | TRCN000 0289833 | |
| Recombinant DNA reagent | LKO scrambled shRNA | Addgene | 1864 | |
| Recombinant DNA reagent | LKO CD169 shRNA | Sigma | TRCN155147 | |
| Recombinant DNA reagent | pLKO.1neo | Addgene | 13425 | |

*Continued on next page*

*Continued*

| Reagent type (species) or resource | Designation | Source or reference | Identifiers | Additional information |
|---|---|---|---|---|
| Recombinant DNA reagent | HIV-1 LAI | PMID: 1683726 | | |
| Recombinant DNA reagent | HIV-1 LAI deltaEnv | PMID: 9245614 | | |
| Recombinant DNA reagent | HIV-1 LAI vpuFS | PMID: 11069982 | | |
| Recombinant DNA reagent | Q23/BG505env | PMID: 29590010; 10364271 | | |
| Recombinant DNA reagent | pHIV-zsGreen | Addgene | 18121 | |
| Recombinant DNA reagent | pHIV-zsGreen/CCR5 | this paper | | Progenitor: pHIV-zsGreen |
| Recombinant DNA reagent | pSMART-LacZ | PMID: 15326157 | | |
| Recombinant DNA reagent | pSCA-helper | PMID: 9660762 | | |
| Chemical compound, drug | Universal Type I Interferon Alpha | PBL Assay Science | 11200–1 | |
| Chemical compound, drug | Puromycin | Sigma | P8833 | |
| Chemical compound, drug | DEAE-Dextran | Sigma | D9885 | |
| Chemical compound, drug | alpha-Chymotrypsin | Sigma | C4129 | |
| Commercial assay or kit | Agencourt AMPure XP Beads | Beckman Coulter | A63880 | |
| Commercial assay or kit | Qubit ds DNA HS Assay Kit | ThermoFisher | Q32854 | |
| Commercial assay or kit | QIAamp viral RNA Kit | Qiagen | 52904 | |
| Commercial assay or kit | One-Step RT-ddPCR Advanced Kit for Probes | BioRad | 1864021 | |
| Commercial assay or kit | Cas9-NLS | UC Berkeley MacroLab | | |
| Commercial assay or kit | Amaxa SG Cell Line 96-well Nucleofector Kit | Lonza | V4SC-3096 | |
| Commercial assay or kit | Epicentre Quick Extract DNA Extraction Solution | Lucigen | QE09050 | |
| Commercial assay or kit | QIAamp DNA Blood Mini Kit | Qiagen | 51185 | |
| Commercial assay or kit | T-20 (Enfuvirtide) | NIH AIDS Reagent Program | 12732 | |
| Commercial assay or kit | HIV-1 p24 ELISA Kit | ABL inc. | 5421 | |
| Software, algorithm | ICE | Synthego (https://ice.synthego.com/#/) | | |
| Software, algorithm | MAGeCK | https://sourceforge.net/p/mageck/wiki/Home/ | PMID: 25476604 | |

## Interferon-Stimulated gene dataset

1905 human ISGs were selected from gene expression datasets of type I IFN-stimulated cells (*Goujon et al., 2013*; *Hung et al., 2015*; *Linsley et al., 2014*) or from previously assembled ISG overexpression (*Schoggins et al., 2011*) or shRNA libraries (*Li et al., 2013*). These included all the genes from the previously assembled ISG libraries (*Li et al., 2013*; *Schoggins et al., 2011*) as well as additional ISGs as defined here. For the GSE46599 dataset (*Goujon et al., 2013*), raw probe-level signal intensities from Illumina HumanHT-12 V4.0 expression BeadChip data were retrieved from GEO, then background-corrected, quantile-normalized and $\log_2$-transformed using the Bioconductor package lumi (*Du et al., 2008*). Fold changes (FC) in expression between type I IFN-treated and untreated samples were calculated for untreated and PMA-treated THP-1 cells, primary CD4+ T cells and primary macrophages. For THP-1 cells, genes with FC $\geq$2 were selected. For primary cells, genes with a donor-specific FC $\geq$2 in at least 2 out of 3 donors were selected. For the GSE60424 dataset (*Linsley et al., 2014*), TMM normalized RNA-seq read count data (Illumina HiScan) were retrieved from GEO. FC in expression in whole blood, isolated CD4 +T cells and monocytes of a Multiple Sclerosis patient, pre- and post-treatment with AVONEX (IFNβ), were calculated and genes with FC $\geq$2 were selected. For the GSE72502 dataset (*Hung et al., 2015*), de novo identification of differentially-expressed genes in IFNα treated PBMCs was performed from the raw RNA sequencing data (Illumina Genome Analyzer). SRA files were retrieved from GEO and converted to FASTQ format using NCBI's SRA toolkit. Reads were mapped to the human reference genome (hg19) using GSNAP (*Wu et al., 2016*) and quantified using HTSeq (*Anders et al., 2015*). Differentially-expressed (DE) genes were identified using the Bioconductor edgeR package (*Robinson et al., 2010*). DE genes were defined at an FDR threshold of 0.05. The glmTreat function was used to detect genes with a FC significantly greater than one between the IFN-treated and control samples. Finally, non-coding RNAs and pseudogenes were removed from the list. Inspection of the curated list of genes showed that overlap between the different datasets was limited and many genes (>2000) were only present in 1 of the 10 datasets/libraries. As such, a second selection round was performed in which the expression threshold for genes present in only one of the datasets was raised to FC $\geq$3. For genes present in at least two datasets, the initial cut-off of FC $\geq$2 was kept. Finally, 35 additional genes identified through RNA sequencing gene expression analysis as being responsive to both type I/type III IFN and IL-1β were also included (M. Gale, personal communication, *Anders et al., 2015*). For analysis of IFN induction specific to THP-1 cells, raw signal intensities were downloaded from GEO (GSE46599) and the data was quantile normalized using the Bioconductor package lumi. For a given probe, both samples from at one least condition were required to have a detection p-value<=0.05. The Bioconductor package limma was used to identify significantly differentially expressed probes. A false discovery rate (FDR) method was employed to correct for multiple testing (*Reiner et al., 2003*), with differential expression defined as |log2 (ratio)|$\geq$0.585 ($\pm$1.5 fold) with the FDR set to 5%.

## Cell culture

The THP-1 monocytic cell line (ATCC; RRID: CVCL_0006) was cultured in RPMI (Invitrogen) with 10% FBS, Pen/Strep, 10 mM HEPES, 0.11 g/L sodium pyruvate, 4.5 g/L D-Glucose and Glutamax. 293T (ATCC CRL-3216; RRID: CVCL_0063) and TZM-bl cells (8129; RRID: CVCL_B478) were cultured in DMEM (Invitrogen) with 10% FBS and Pen/Strep. For some validation studies, THP-1 cells with single-cell sorted into 96-well plates to create individual clonal lines (BD FACS Aria II – Fred Hutch Flow Cytometry Core). Universal Type I Interferon Alpha was obtained from PBL Assay Science (Catalog No. 11200–2), diluted to $10^5$ Units/mL in sterile-filtered PBS/1% BSA according to the activity reported by manufacturer and frozen in aliquots at −80°C. All Puromycin selections were done at 0.5–1 ug/mL. The identity of THP-1 cells was confirmed by STR profiling (Fred Hutch Research Cell Bank). Mycoplasma was detected in some THP-1 cultures after the completion of experiments.

## Plasmids

lentiCRISPRv2 plasmid was a gift from Feng Zhang (Addgene #52961). pMD2.G and psPAX2 were gifts from Didier Trono (Addgene #12259/12260). lentiCRISPRv2 constructs targeting genes of interest were cloned into BsmBI-digested lentiCRISPRv2 by annealing complementary oligos (*Supplementary file 1*) with overhangs that allow directional cloning into lentiCRISPRv2. Stable LKO

SEC62 shRNA lentiviral vectors were obtained from Sigma. SEC62_1: CCGGCCAGGAAATCA TGGAACAGAACTCGAGTTCTGTTCCATGATTTCCTGGTTTTTG (TRCN0000289739). SEC62_2: CCGGGAAATGAGAGTAGGTGTTTATCTCGAGATAAACACCTACTCTCATTTCTTTTTG (TRCN0000289833). Scramble shRNA (CCTAAGGTTAAGTCGCCCTCGCTCGAGCGAGGGCGAC TTAACCTTAGG) was a gift from David Sabatini (Addgene #1864). The CD169 shRNA (Sigma TRCN155147) (CCGGGTGTGGAGATTCACAACCCTTCTCGAGAAGGGTTGTGAATCTCCACAC TTTTTTG) was a gift from Rahm Gummuluru. The CD169 shRNA sequence was subcloned into pLKO.1neo (Addgene #13425) using EcoRI and AgeI sites. HIV-CRISPR was constructed (Genscript) by inserting a synthesized 433 bp sequence from HIV-1$_{LAI}$ into the deleted 3' LTR U3 sequence of lentiCRISPRv2. HIV-1$_{LAI}$ LT insert: ATCCTTGATCTGTGGATCTACCACACACAAGGCTACTTCCCTGA TTGGCAGAACTACACACCAGGGCCAGGGGTCAGATATCCACTGACCTTTGGATGGTGC TACAAGCTAGTACCAGTTGAGCCAGATAAGGTAGAAGAGGCCAATAAAGGAGAGAACACCAGC TTGTTACACCCTGTGAGCCTGCATGGAATGGATGACCCTGAGAGAGAAGTGTTAGAGTGGAGG TTTGACAGCCGCCTAGCATTTCATCACGTGGCCCGAGAGCTGCATCCGGAGTACTTCAAGAAC TGCTGACATCGAGCTTGCTACAAGGGACTTTCCGCTGGGGACTTTCCAGGGAGGCGTGGCC TGGGCGGGACTGGGGAGTGGCGAGCCCTCAGATGCTGCATATAAGCAGCTGCTTTTTGCCTG TACTGGGTCTCTCTGGTTA. The wild type (HIV-1$_{LAI}$), *env*-deleted (HIV-1$_{LAI}$ VSV-G) and *vpu*-deficient (HIV$_{LAI}$Δvpu = VpuFS/Rap5) HIV-1$_{LAI}$ proviruses were previously described (*Bartz and Vodicka, 1997*; *Gummuluru et al., 2000*; *Peden et al., 1991*). The Clade A HIV-1$_{Q23.BG505}$ molecular clone was previously described (*Haddox et al., 2018*; *Poss and Overbaugh, 1999*). The pHIV-zsGreen/ CCR5 construct was created by cloning the human CCR5 CDS into pHIV-zsGreen (Addgene #18121) with BamHI and NotI.

## ISG CRISPR/Cas9 sgRNA Library Construction

Four sgRNA sequences were selected randomly from the Brunello library for each gene target (*Doench et al., 2016*) and four additional non-identical sgRNAs were subsequently selected randomly from the Genome-scale CRISPR Knock-Out (GeCKO v2) library (*Sanjana et al., 2014b*; *Shalem et al., 2014a*). For genes for which eight unique sgRNAs could not be obtained from these libraries, additional sgRNAs were added from the Moffat (*Hart et al., 2015*) and Sabatini/Lander libraries (*Wang et al., 2015*; *Wang et al., 2014*). 12 genes contained no sgRNAs in any of the libraries and for those genes eight new sgRNAs were designed using the sgRNA *Designer* from the *Broad Institute* (http://portals.broadinstitute.org/gpp/public/analysis-tools/sgrna-design). A total of 15,348 unique sgRNA sequences were synthesized. The sgRNAs were split in two pools (Zhang and Brunello sgRNAs) for synthesis (four per gene in each pool) and two independent sets of 200 Non-Targeting Control (NTC) sgRNAs obtained from the GeCKOv2 library were added in duplicate to each pool. The PIKA$_{HIV}$ ISG-targeting sgRNA library was synthesized (Twist Biosciences) and cloned into HIV-CRISPR. Oligo pools were amplified using Phusion HF (Thermo) using 1 ng of pooled oligo template, primers ArrayF and ArrayR (ArrayF primer: TAACTTGAAAGTATTTCGATTTCTTGGCTTTATATATC TTGTGGAAAGGACGAAACACCG and ArrayR primer: ACTTTTTCAAGTTGATAACGGACTAGCC TTATTTTAACTTGCTATTTCT AGCTCTAAAAC), an annealing temperature of 59°C, an extension time of 20 s, and 25 cycles. Following PCR amplification, a 140 bp amplicon was gel-purified and cloned into BsmBI digested vectors using Gibson assembly (NEB). Each Gibson reaction was carried out at 50°C for 60 min in a thermocycler. 1 μl of the reaction was used to transform 25 μl of electro-competent cells (Stellar Competent Cells; Clontech) according to the manufacturer's protocol using a GenePulser (BioRad). To ensure adequate representation, sufficient parallel transformations were performed and plated onto ampicillin containing LB agarose 245 mm x 245 mm plates (Thermo Fisher) at 200-times the total number of oligos of each library pool. After overnight growth at 37°C, colonies were scraped off, pelleted, and used for plasmid DNA preps using the Endotoxin-Free Nucleobond Plasmid Midiprep kit (Takara Bio #740422.10). The PIKA$_{HIV}$ plasmid library was sequenced and contains 15,339 of the 15,348 total sgRNAs included in the synthesis (*Figure 1—figure supplement 2*).

## Virus and lentivirus production

293 T cells (ATCC) were plated at $2 \times 10^5$ cells/mL in 2 mL in 6-well plates one day prior to transfection using TransIT-LT1 reagent (Mirus Bio LLC) with 3 μL of transfection reagent per μg of DNA. For

lentiviral preps, 293Ts were transfected with 667 ng lentiviral plasmid, 500 ng psPAX2 and 333 ng MD2G. For HIV-1 production, 293Ts were transfected with 1 ug/well proviral DNA. One day post-transfection media was replaced. Two- or three- days post-transfection viral supernatants were clarified by centrifugation (1000 g) and filtered through a 20 µm filter. For PIKA$_{HIV}$ library preps, supernatants from 40 × 6 well plates were combined and concentrated by ultracentrifugation. 30 mL of supernatant per SW-28 tube were underlaid with sterile-filtered 20% sucrose (1 mM EDTA, 20 mM HEPES, 100 mM NaCl, 20% sucrose) and spun in an SW28 rotor at 23,000 rpm for 1 hr at 4°C in a Beckman Coulter Optima L-90K Ultracentrifuge. Supernatants were decanted, pellets resuspended in DMEM over several hours at 4°C and aliquots frozen at −80°C. All viral and lentiviral infections and transductions were done in the presence of 20 µg/mL DEAE-Dextran (Sigma; D9885). Semliki Forest Virus (SFV) stocks were created by co-transfection of SFV replicon (pSMART-LacZ) and helper (pSCA-helper) plasmids in 293Ts as previously described (*DiCiommo and Bremner, 1998*; *DiCiommo et al., 2004*). SFV particles were harvested 48 hr post-transfection and activated by treatment with α-Chymotrypsin (Sigma #C4129).

## PIKA$_{HIV}$screening

Large-scale preps of the PIKA$_{HIV}$ lentiviral library were titered by a colony-forming assay in TZMbl cells and used to transduce THP-1 cells at an MOI of 0.7. Cells were selected in Puromycin (0.5 µg/mL) for 10–14 days. 8 × 10$^6$ cells per replicate (>500X coverage of the PIKA$_{HIV}$ library) were treated or not with IFNα and infected the following day at a viral dose determined to allow approximately 50% of cells in culture to be infected by spinoculation at 1100xg for 30 min with 20 µg/mL DEAE-Dextran. After overnight incubation, cells were resuspended in media with or without IFNα at 5 × 10$^5$ cells/mL. Cells and supernatants were collected 3 days post infection. Genomic DNA was extracted from cell pellets with a QIAamp DNA Blood Midi Kit (Qiagen #51183) and genomic DNA eluted in water. Viral supernatants were spun at 1100xg to remove cell debris, filtered through a 0.2 µm filter, overlaid on a 20% sucrose cushion and concentrated in SW28 rotor for 1 hr at 4°C. After resuspension in PBS, viral RNA was extracted (QIAamp viral RNA Kit, Qiagen, 52904). sgRNA sequences present in the genomic DNA and viral supernatants were amplified by PCR and RT-PCR, respectively, using primers specific for the HIV-CRISPR construct (*Supplementary file 1*) (*Toledo et al., 2015*). Libraries were then barcoded/prepared for Illumina sequencing by a second round of PCR (*Supplementary file 1*). Each amplicon was then cleaned up through double-sided SPRI (Agencourt AMPure XP Beads – Beckman Coulter #A63880), quantitated with a Qubit dsDNA HS Assay Kit (Q32854 – ThermoFisher) and pooled to 2 nm for each library. Pooled, multiplexed libraries were then sequenced on a single lane of an Illumina HiSeq 2500 in Rapid Run mode (Fred Hutch Genomics and Bioinformatics Shared Resource).

## Screen analysis

Raw sequence read data from screens are available as a GEO DataSet #GSE118631. Following demultiplexing of libraries to assign sequences to each sample (allowing no mismatches), reads were trimmed and aligned to the PIKA$_{HIV}$ sgRNA library, using Bowtie (*Langmead et al., 2009*). NTC sgRNA sequences were iteratively binned to create an NTC sgRNA set as large as the ISG gene set in the PIKA$_{HIV}$ library. Relative enrichment or depletion of sgRNAs and genes were analyzed using the MAGeCK statistical package (*Li et al., 2014*). For the VSV-G screen a single IFITM1-targeting sgRNA sequence (AGCATTCGCCTACTCCGTGA) with complete homology to IFITM3 was removed from the analysis.

## Sliding window analysis of CG dinucleotide content

An Excel (Microsoft) worksheet was created to analyze the CG dinucleotide content of the HIV-CRISPR-NTC1 transcript from the beginning of the 5′R region to the end of the 3′R region. The HIV-CRISPR sequence was broken into fragments of 3 nucleotides (codons), and at each position the number of CG dinucleotides within or between two adjacent codons was determined. The CG counts at each position over the length of the sequence were then summed within a sliding window of 67 codons (201 nucleotides) and plotted against the nucleotide position of the transcript in GraphPad Prism.

## Digital droplet PCR (ddPCR)

Wild-type or ZAP-knockout THP-1 cells were transduced with a pooled library of HIV-CRISPR encoding 39 distinct gRNAs at an MOI of 0.5 and selected with puromycin for 15 days as described above. Cells were infected with HIV-1$_{LAI}$ at an MOI of 1. Three days post-infection, viral supernatants were cleared by centrifugation, filtered through a 0.4 µm filter, and viral RNA was extracted from 140 µL of supernatant using the QIAamp viral RNA Kit, with subsequent aliquoting and freezing at −80$^0$C. cDNA was synthesized from viral RNA with random hexamers (OneStep RT-ddPCR Advanced Kit for Probes, BioRad, 1864021), and the number of copies of either HIV-1$_{LAI}$ or HIV-CRISPR genomes per µL of cDNA was quantified by ddPCR using the QX200 Droplet Digital PCR System (Bio-Rad, Hercules, CA). HIV was detected using previously published primers and probe directed towards *pol* (*Benki et al., 2006*). To specifically detect HIV-CRISPR, we used primers ddPCR-cPPT-F (GTA CAG TGC AGG GGA AAG), ddPCR-U6-R (ATG GGA AAT AGG CCC TCG), and probe cPPT-probe (6-FAM/ZEN- AGA CAT AAT AGC AAC AGA CAT ACA AAC -IBFQ) (Integrated DNA Technologies, Skokie, IL). Both sets of reactions were set up according to the manufacturer's protocols with an annealing temperature of 60$^0$C. The HIV-CRISPR primers were found to be specific, as no amplification was detected in untransduced cells infected with HIV-1. Control reactions on viral RNA without reverse transcriptase revealed that carry-over plasmid contamination from viral preps accounted for only a low level (<50 copies/µL) of amplification.

## Knockout, Knockdown and Overexpression Cell Pools and Clones

ZAP knockout cell pools were created by electroporating THP-1 cells with a custom ZAP-targeting crRNA (ATGTGGAGTCTTGAACACGG; IDT). 1 µL crRNA was resuspended at 160 µM in 10 mM Tris pH 7.4 and complexed at an equimolar ratio with 1 µL 160 µM tracrRNA (IDT #1072534) and incubated 30 min at 37°C followed by addition of 2 µL of 40 µM Cas9-NLS (UC Berkeley MacroLab) and further incubation at 37°C for 15 min to create the ZAP-targeting crRNP complexes. 3.5 µL crRNP was added to 5 × 10$^5$ THP-1 cells resuspended in Amaxa SG Cell Line 96-well Nucleofector Kit (Lonza #V4SC-3096) and electroporated according to the manufacturer's protocol (Lonza 4D Nucleofector). 80 µL of prewarmed media was added, followed by incubation for 30 min recovery in the 37°C incubator. Cells were then resuspended at 2.5 × 10$^5$ cells/mL in 500 µL in a 24-well plate for 48 hr before single cell sorting into 96 well U-bottom plates containing RPMI media supplemented with 20% FBS (BD FACS Aria II – Fred Hutch Flow Cytometry Core). MxB-KO clonal lines were generated by transduction with lentiCRISPRv2 containing MxB-targeting sgRNA sequences (see Supplemental Table S5 for sgRNA sequences) followed by single-cell cloning and puromycin selection. lentiCRISPRv2 KO cell pools targeting N4BP1, STAT1, STAT2, IRF9, MxB, TRIM5alpha, IFITM1, Tetherin, CXCR4 and TLR2 as well as two Non-Targeting Controls (NTC_1 and NTC_2) were created through transduction with lentivirus and selection in Puromycin (see Supplemental Table S5 for sgRNA sequences). Both KO cell pools and individual KO cell lines were validated using Western blotting, flow cytometry and/or genomic editing analysis as described below. shRNA knockdown cell pools were made by transducing wildtype THP-1 cells with lentiCRISPRv2 shRNA constructs and selected in RPMI containing 1 µg/mL Puromycin for two weeks prior to validating via Western blotting or flow cytometry. CCR5-overexpressing ZAP-KO THP-1 cells were created by transducing ZAP-KO THP-1 cells with pHIV-zsGreen/CCR5 lentiviral supernatants and sorting a zsGreen +population 48 hr post-transduction.

## Genomic editing analysis

Knockout cells were harvested and either lysed in Epicentre QuickExtract DNA Extraction Solution (Lucigen QE09050) for direct PCR amplification or genomic DNA was extracted (QIAamp DNA Blood Mini Kit – Qiagen #51185). Edited loci were amplified from cell pool DNA using primers specific to each targeted locus (*Supplementary file 1*) as previously published (*Hultquist et al., 2016*). PCR amplicons were sequenced (Fred Hutch Shared Resources Genomics Core – sanger sequencing) and analyzed by ICE (Synthego) to determine the percent of alleles edited at each locus in the cell population (*Hsiau et al., 2018*). Editing was confirmed at each locus (*Figure 4—source data 1*; KO scores varied across pools from 48% to 93%).

## Antibodies

For Western blotting the following antibodies were used as follows: MxB (Santa Cruz sc-271527; RRID: AB_10649506) at 1:200, Sec62 (Abcam ab168843) at 1:2000, ZAP/ZC3HAV1 (Proteintech #16820–1-AP; RRID: AB_2728733) at 1:5000, N4BP1 (Cohesion Biosciences #CPA2415) at 1:1000, tubulin (Sigma T6199; RRID: AB_477583) at 1:1000 and actin (Sigma A2066; RRID: AB_476693) at 1:5000. Secondary antibodies were used as follows: 1:5000 donkey anti-goat IgG-HRP (Santa Cruz Biotechnology sc-2020; RRID: AB_631728) and 1:5000 goat anti-rabbit IgG-HRP (Santa Cruz Biotechnology sc-2004; RRID: AB_631746). For flow cytometry, antibodies were used as follows: CD4 (BD Pharmingen 555349; RRID: AB_398593) 1:50, CXCR4 (eBioscience 17-9999-42; RRID: AB_1724113) 1:50, CD-169 (BioLegend 346003; RRID: AB_2189038) 1:50, TLR2 (BioLegend 309707; RRID: AB_314777) 1:100, Tetherin (BioLegend 348410; RRID: AB_2067121) 1:50.

## Flow cytometry

For intracellular $Gag_{p24}$ (p24) staining, cells were harvested and fixed in 4% paraformaldehyde for 10 min and diluted to 1% in PBS. Cells were permeabilized in 0.5% Triton-X for 10 min and stained with 1:300 KC57-FITC (Beckman Coulter 6604665; RRID: AB_1575987). Cells were read on a BD FACS-CANTO II (Fred Hutch Flow Cytometry Core) and analyzed in FlowJo. For cell surface marker staining, cells were washed twice in PBS, stained in PBS/1% BSA, incubated at 4°C for 1 hr, washed twice in PBS, and analyzed on the Canto two flow cytometer (Fred Hutch Flow Cytometry Core).

## Western blotting

Cells were lysed in 2X SDS-SB lysis buffer (10% glycerol, 2% BME, 6% SDS, 62.5 mM Tris-HCl pH 6.8), boiled at 95*C, sonicated for one minute and resolved by NuPAGE 4–12% Bis-Tris Gel (Invitrogen). Following transfer to a PVDF membrane and blocking in PBS/5%milk for 1 hr, blots were probed with antibodies for 1 hr or overnight, washed in PBST, probed with HRP secondary, washed in PBST and bands visualized with SuperSignal West Femto Maximum Sensitivity Substrate (Thermo-Fisher #34095). Blots were visualized on a BioRad Chemidoc MP.

## Viral infectivity assays

Cells were pre-stimulated with IFNα 24 hr prior to infection. Virus and 20 µg/mL DEAE-Dextran in RPMI were added to cells, spinoculated for 20 min at 1100x*g*, and incubated overnight at 37°C. Cells were washed the next day and re-suspended in RPMI supplemented with IFNα. For experiments to assay ISGs affecting late steps in viral replication, cells were spinoculated at 1100x*g* for 20 min with HIV-1$_{LAI}$ or Vpu-deficient HIV-1$_{LAI}$ (HIV$_{LAI}\Delta vpu$) at an MOI of 0.4, incubated at 37°C for 16 hr, and then treated with 1000 mU/mL IFNα. 24 hr post infection, cells were washed of virus and re-suspended in interferon containing media with 1 µg/mL T-20 entry inhibitor (NIH AIDS Reagent Program, Division of AIDS, NIAID, NIH: Enfuvirtide #12732). For SFV infections THPs were infected overnight with SFV-LacZ stocks and LacZ activity assayed the following day with the Gal-Screen β-Galactosidase Reporter Gene Assay System (ThermoFisher #T1029).

## Virus release (p24 ELISA and RT assay)

p24 ELISA on cell culture supernatants was performed with a HIV-1 p24 Ag Assay (ABL #5421). Reverse transcriptase activity in viral supernatants was measured using the RT activity assay as described (*Roesch et al., 2018*; *Vermeire et al., 2012*). A stock of HIV-1$_{LAI}$ virus was titered multiple times, aliquoted at −80°C and used as the standard curve in all assays.

## Acknowledgements

We thank Patrick Paddison, Phil Corrin, Lucas Carter, Yu Ding, Pia Hoellerbauer, Dan Kuppers for assembly of the PIKA library and assistance with screening methodology, Stephanie Rainwater and Abby Felton for technical assistance, Chris Large for a parallel screen using the PIKA library for genes necessary for induction of ISGs and the Fred Hutch Shared Resources Bioinformatics and Genomics Cores (NCI 5 P30 CA015704-43) and Harmit Malik for discussions and comments on the manuscript. This work was supported by NIH grant R01 AI30927 (ME), a CCEH Pilot Grant P30 DK56465 (MO), UW/FHCRC CFAR New Investigator Award P30 AI027757 (MO) and a Belgian

American Educational Foundation Fellowship (JV). Construction of the PIKA library was supported by DP1 DA039543 to Julie Overbaugh.

## Additional information

### Funding

| Funder | Grant reference number | Author |
|--------|------------------------|--------|
| National Institute of Allergy and Infectious Diseases | CFAR New Investigator Award, P30 AI027757 | Molly OhAinle |
| National Cancer Institute | P30 CA015704-43 | Jeffrey J Delrow |
| National Institute of Diabetes and Digestive and Kidney Diseases | CCEH Pilot Grant, DK56465 | Molly OhAinle |
| Belgian American Educational Foundation | Postdoctoral Fellowship | Jolien Vermeire |
| National Institute on Drug Abuse | DP1 DA039543 | Julie Overbaugh |
| National Institute of Allergy and Infectious Diseases | R01 AI30927 | Michael Emerman |

The funders had no role in study design, data collection and interpretation, or the decision to submit the work for publication.

### Author contributions

Molly OhAinle, Conceptualization, Data curation, Formal analysis, Supervision, Funding acquisition, Validation, Investigation, Visualization, Methodology, Writing—original draft, Project administration, Writing—review and editing; Louisa Helms, Data curation, Formal analysis, Validation, Investigation, Writing—review and editing; Jolien Vermeire, Conceptualization, Supervision, Investigation, Methodology, Writing—review and editing; Ferdinand Roesch, Methodology, Writing—review and editing; Daryl Humes, Formal analysis, Validation, Investigation, Writing—review and editing; Ryan Basom, Data curation, Formal analysis, Visualization, Methodology, Writing—review and editing; Jeffrey J Delrow, Supervision, Methodology; Julie Overbaugh, Resources, Funding acquisition, Writing—review and editing; Michael Emerman, Conceptualization, Resources, Supervision, Funding acquisition, Project administration, Writing—review and editing

### Author ORCIDs

Molly OhAinle (iD) http://orcid.org/0000-0001-5138-5367
Michael Emerman (iD) https://orcid.org/0000-0002-4181-6335

### Decision letter and Author response

Decision letter https://doi.org/10.7554/eLife.39823.033
Author response https://doi.org/10.7554/eLife.39823.034

## Additional files

### Supplementary files

• Supplementary file 1. Oligos and Primers. Tab 1 (sgRNA oligos): oligo name. oligo_seq. sgRNA name. seq. ICE_F oligo. ICE_R oligo. Tab 2 (sequencing primers): oligo_name. sequence.
DOI: https://doi.org/10.7554/eLife.39823.022

• Transparent reporting form
DOI: https://doi.org/10.7554/eLife.39823.023

## Data availability

Sequence data generated for this study is available at the NCBI Gene Expression Omnibus (GEO) under accession number GSE118631. All data generated are included in the manuscript and supporting files. Source data files have been provided.

The following dataset was generated:

| Author(s) | Year | Dataset title | Dataset URL | Database and Identifier |
|---|---|---|---|---|
| Molly OhAinle, Jolien Vermeire, Ferdinand Roesch, Daryl Humes, Ryan Basom, Jeffrey J Delrow, Julie Overbaugh, Michael Emerman, Louisa Helms | 2018 | A Virus-Packageable CRISPR Screen Identifies Host Factors Mediating Interferon Inhibition of HIV | https://www.ncbi.nlm.nih.gov/geo/query/acc.cgi?acc=GSE118631 | NCBI Gene Expression Omnibus, GSE118631 |

The following previously published datasets were used:

| Author(s) | Year | Dataset title | Dataset URL | Database and Identifier |
|---|---|---|---|---|
| Goujon C, Schulz R, Mirza M, Malim MH | 2013 | Genome-wide analysis of interferon-stimulated genes in primary cells and immortalized cell lines | https://www.ncbi.nlm.nih.gov/geo/query/acc.cgi?acc=GSE46599 | NCBI Gene Expression Omnibus, GSE46599 |
| Speake C, Linsley PS, Whalen E, Chaussabel D, Presnell SR, Mason MJ, Gersuk VH, O'Brien KK, Nguyen Q, Greenbaum CJ, Buckner JH, Malhotra U | 2015 | Next generation sequencing of human immune cell subsets across diseases | https://www.ncbi.nlm.nih.gov/geo/query/acc.cgi?acc=GSE60424 | NCBI Gene Expression Omnibus, GSE60424 |
| Hung T, Behrens T, Chaivorapol C, Ortmann W | 2015 | Healthy donor PBMC RNA-seq with or without interferon-alpha stimulation | https://www.ncbi.nlm.nih.gov/geo/query/acc.cgi?acc=GSE72502 | NCBI Gene Expression Omnibus, GSE72502 |

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
