## [Decision Letter]

[**Editorial note:** This article has been through an editorial process in which the authors decide how to respond to the issues raised during peer review. The Reviewing Editor's assessment is that all the issues have been addressed.]

Acceptance letter:

All the significant issues have been addressed. The major issue raised was that the reviewers felt it was important for the authors to validate that their CRISPR “knockouts” had actually worked. It turned out that the authors had actually done this, and they clarified this point in their revised manuscript. The authors also added a number of other improvements in the revised manuscript, including repeating the CRISPR screen on another (R5-tropic) HIV-1 strain and clarified a series of minor issues in response to reviewer questions.

Revision letter:

Thank you for submitting your article "A Virus-Packageable CRISPR Screen Identifies Host Factors Mediating Interferon Inhibition of HIV" for consideration by *eLife*. Your article has been reviewed by three peer reviewers, and the evaluation has been overseen by a Reviewing Editor and Michel Nussenzweig as the Senior Editor. The reviewers have opted to remain anonymous.

The Reviewing Editor has highlighted the concerns that require revision and/or responses, and we have included the separate reviews below for your consideration. If you have any questions, please do not hesitate to contact us.

Summary:

Ohainle et al. describe an elegant CRISPR/Cas9-based screening strategy for identifying host factors that regulate the HIV life cycle. The screen's design is clever, based on modified HIV-based sgRNA lentivectors that allow them to track, using deep sequencing, sgRNAs encapsidated into virus particles after infection of the knock-out pool with replication-competent HIV-1. As the CRISPR vector can be mobilized by replication-competent HIV-1 replication in the same cell, sgRNA-CAS9 vectors that target putative antiviral genes will be overrepresented in the supernatant of infected cells pretreated with IFN. The screen includes key controls that should be screened out (e.g., HIV entry factors and those involved in the transduction of IFN signaling), and, importantly, they were. The screen revealed a number of ISGs previously shown to inhibit retroviral replication. Of these, ZAP, TRIM25 and N4BP1 appear to be targeting the CRISPR vector rather than HIV itself, most likely owing to the high CpG content of the CAS9-P2A-PuroR mRNA/vector genome RNA. For the other ISGs identified, the authors show that hTRIM5, tetherin, MxB and several IFITMs contribute to the IFN-induced inhibition of HIV replication. The screening strategy was validated by studying the effects of interferon in conferring antiviral phenotypes in THP-1 monocytes. Finally, by examining sgRNAs that are under-represented, the authors identified SEC62 and TLR2 as putative viral cofactors in THP-1 cells. Given that the sgRNA library did not cover the whole genome, it seems likely that many more host cofactors could be identified using this experimental approach.

Overall, this is a fascinating project, very nicely performed, and the majority of paper is clearly laid out and well-written. While the major contribution is the derivation and optimization of a powerful new screening method for identifying HIV-host interactions, validation of the screen also yields interesting new restriction and host dependency factors impacting the HIV life cycle and insights regarding the multi-layer nature of cellular antiviral immunity. A particularly intriguing finding is the demonstration that human versions of antiviral host proteins typically thought to be circumvented by HIV-1 (e.g., TRIM5a and BST2/Tetherin) retain a significant level of anti-HIV activity in IFN-treated THP-1 cells. However, the new HIV biology could be enhanced by increased focus on their mechanisms, at least in the Discussion (e.g., the identification of human TRIM5alpha in this screen was unexpected, but no attempt was made to clarify how it functions, which is of interest given that it has largely been regarded by the field as an ineffective inhibitor of HIV-1). Nevertheless, most of the paper's weaknesses are minor and largely pertain to clarifications that could be made to the description and interpretation of a subset of experiments.

Major Issue:

At least a subset of the knock-out populations should be validated experimentally (e.g., by western blot, endonuclease or sequencing assays). Similarly, in no instances were the KO gene activities rescued to confirm KO specificity. The authors might at least comment on why they felt this was unnecessary.

Minor Issues:

1) In Figure 1, the rationale for repairing the LTR to generate packageable RNA species could be more clearly described for a general audience. Also, is there potential for promoter interference in this context?

2) Why was cellular gDNA was sequenced and not cell-associated RNA? While gDNA would give a representation of each guide RNA as integrated into the cell pool, it does not confirm that the guide RNAs are expressed. Could this cloud the analysis? A related question is whether essential genes that may be involved in the viral life cycle can be detected in this screen based on loss of sequences? Finally, how was ZAP still identified as a mid-level hit even in ZAP KO cells (subsection “An iterative PIKAHIV screen in ZAP-KO cells identifies a panel of ISGs that inhibit HIV in THP-1 cells”, second paragraph)?

3) N4BP1 could be highlighted more strongly (e.g., in Abstract) considering that it appears to be a new potential regulator of the ZAP pathway.

4) For Figure 2C, it could be better explained why it makes sense that gene correlations are stronger for the ZAP-KO cells than for wild-type cells.

5) For Figure 3 results, please clarify the rationale for testing entry pathways (as laid out in the context of MxB). As noted, the authors have already published that VSV-G is more sensitive to IFITM proteins. The data seem to indicate that the MxB restriction is 5 to 10-fold for either context, so it is unclear how the entry pathway was relevant to MxB restriction.

6) For Figure 4E results, for a general audience, the authors need to define in the text what Vpu does (e.g., subsection “TRIM5alpha, IFITM1 and Tetherin are additional ISGs that contribute to the IFN block”, last paragraph).

7) The authors mention "genetic conflicts" on several occasions as a possible explanation for why the virus is unable to completely antagonize human RFs such as TRIM5alpha and BST2, without always explaining this concept very clearly. For example, a major conclusion is that T5alpha, etc. affects "well-adapted" HIV-1-LAI (Discussion, first paragraph), but what does "well-adapted" mean? They mention "conflicting evolutionary pressures" acting on the genome but only much later in the Discussion do they begin to explain what they mean by this (e.g., that evasion might come at a cost for other viral functions). They also mention "concurrently-acting RF barriers", but this potential explanation should be clarified.

8) For the Discussion, we suggest addressing the question of how much ZAP might limit the application of this screening strategy for primary cells wherein it would be difficult and perhaps less-than-ideal to inactivate ZAP directly. Instead, would it be feasible to remove CpG dinucleotides from the library? As noted above, there could also be more emphasis on the data that nicely implicates both TRIM25 and N4BP1 as downstream regulators of ZAP restriction.

9) The comparison of this screen to that of Park et al. should be a separate paragraph in the Discussion, and more could be said here, both to explain the Park screen better and articulate the pros and cons of either screen and their findings.

10) APOBECs are brought up in the Discussion and are apparently not detected in this screen due to its "single-round" nature. Would it be possible to modify the screen wherein it could detect APOBEC-like activities (i.e., impacting RT, integrase, or some other aspect of multi-round replication)? This could be discussed.

11) The screen in THP-1 is limited to X4-using or VSV-G pseudotyped HIV-1, using the lab-adapted prototypic strain LAI. It would be interesting to see whether the same genes are identified if a primary HIV-1 isolate was used instead. This would either require challenging the cells with a VSV-G pseudotype or making the THP-1 ZAP-KO cells CCR5 positive. The authors touch on the idea that the lab-adapted strain may be have residual sensitivity to certain ISGs, and there is good evidence of a spectrum of IFN sensitivities amongst primary HIV isolates. The system the authors have developed gives them the opportunity to do interesting comparative studies which will yield further novel observations. The authors should at least discuss this more fully if they feel re-running the screen with a different virus/virus mutant is beyond the scope.

12) Similarly, the relative impact of MxB vs. IFITM restrictions depending on route of entry would be strengthened by comparing other HIV-1 envelopes, given other publications in the field. Of note, a very closely related molecular clone derived from NL/HxB, but not an Env-deleted VSV pseudoype of the same virus, has been shown to be completely insensitive to IFNa in the THP-1 cells despite robust Mx2 induction (Rihn et al., JVI 2017).

13) Note that the identification of tetherin in the screen could be because the Vpu encoded LAI/IIIB lineage is a sub-optimal tetherin antagonist. While the authors allude to compromised Vpu function in the Discussion, they should refer to Pickering et al., 2014, where essentially the same Vpu allele (NL4.3) was specifically shown to be an inferior tetherin antagonist compared to most primary isolates tested.

14) As noted above, the manuscript would benefit from additional mechanistic information (or at least increased discussion) for several of the more interesting "hits". In particular, identification of human TRIM5 is somewhat surprising considering that HIV-1 is supposed to be largely resistant. How much does TRIM5 contribute to the enhanced IFN-sensitivity of certain capsid mutants? Is TRIM5 contributing to IFN sensitivity of other HIV-1 isolates in the same system? It is suggested that the TRIM5 effects could reflect induced expression as an ISG, but this was not addressed. Similarly, TRIM5alpha restriction was bypassed by VSVg pseudotyping (Figure 3E): is this a general feature of TRIM5 restriction, or is it a characteristic of this experimental set-up and/or THP cells? Likewise, there is preliminary evidence that N4BP1 could be a ZAP cofactor, but this is not further characterized. Finally, the identified novel dependency factors could be characterized more fully. The identification of TLR2 and (to a lesser extent) its adaptor MyD88 as dependency factors suggests downstream signaling. Do the authors have any theories regarding TLR2 as a dependency factor? Do they believe this effect is via signaling, and if it is, what the agonist may be? Can this phenotype be explained by knockout affecting NFkB activity and therefore the activity of the viral LTR (of note NFkB1 is also a hit)? As there is no TLR2 ligand added to the culture, what could be stimulating the system here? SEC62 appears to be affecting CD4, but not CXCR4, surface expression and this correlates with sensitivity of LAIwt but not LAI(VSV). Are other HIV-1 envelopes that require different cell densities if CD4 more or less sensitive to SEC62 knockout?

15) Did the authors undertake a deep sequencing analysis of the PIKA-HIV library prior to generating the transducing virus stock in 293T cells? It would be useful to know if the transduced THP population already carried some bias in its sgRNA content (for instance, it might be anticipated that guides recognizing dependency factors affecting the late stage of virus replication would be under represented).

Text edits:

1) Much of the paper is written in present tense.

2). For a broad audience, be sure to define abbreviations and jargon in Materials and methods and elsewhere, and also other esoteric terms (e.g., "massively-parallel").

3) Introduction, second paragraph; might cite original papers or more recent reviews, for example MxB not reported until 2013.

4) "MxB is a the"; "a" vs. "the" will make a big difference in this sentence.

5) "these pseudotyped", rather "of VSV-G pseudotyped"?

6) "may due".

7) Subsections “Cell Culture”, “Virus and lentivirus production”, and “Knockout and Knockdown Cell Pools and Clones”, some mis-capitalizations.

Additional data files and statistical comments:

Rigor and statistics seemed adequate.

The separate reviews are included for reference:

*Reviewer #1:*

Ohainle et al. describe an elegant CRISPR/Cas9-based screening strategy for identifying host factors regulating the HIV life cycle. The screen's design is clever, based on modified HIV-based sgRNA lentivectors that allow them to track, using deep sequencing, sgRNAs encapsidated into virus particles after infection of the knock-out pool with replication-competent HIV-1. Herein, relevant sgRNAs were identified based on enrichment in or loss from HIV virus particles after single-round infection. The screening strategy was validated by studying the effects of interferon in conferring antiviral phenotypes in THP-1 monocytes.

Overall, this is a fascinating project very nicely performed, and the majority of paper is clearly laid out and well-written. While the major contribution is the derivation and optimization of a the powerful new screening method for identifying HIV-host interactions, validation of the screen also yields interesting new restriction and host dependency factors impacting the HIV life cycle and insights regarding the multi-layer nature of cellular antiviral immunity. An intriguing finding is demonstration that human versions of antiviral host proteins typically thought to be circumvented by HIV-1 (e.g., TRIM5a and BST2/Tetherin) retain a significant level of anti-HIV activity in IFN-treated THP-1 cells. The paper's weaknesses are minor and pertain to clarifications that could be made to the description and interpretation of a subset of experiments, as itemized in minor comments.

Minor Comments:

1) In Figure 1, the rationale for repairing the LTR to generate packageable RNA species could be more clearly described for a general audience. I was also curious if there is potential for promoter interference in this context?

2) For the screen, I was also curious as to why cellular gDNA was sequenced and not cell-associated RNA? While gDNA would give a representation of each guide RNA as integrated into the cell pool, it does not confirm that the guide RNAs are expressed. Could this cloud the analysis? A second, related question was whether or not essential genes that may or may not be involved in the viral life cycle can be detected in this screen based on loss of sequences? Finally, how was ZAP still identified as a mid-level hit even in ZAP KO cells (subsection “An iterative PIKAHIV screen in ZAP-KO cells identifies a panel of ISGs that inhibit HIV in THP-1 cells”, second paragraph)?

3) N4BP1 could be highlighted more strongly (e.g., in Abstract) considering that it appears to be a new potential regulator of the ZAP pathway.

4) For Figure 2C, it could be better explained why it makes sense that gene correlations are stronger for the ZAP-KO cells than for wild-type cells.

5) For Figure 3 results, I found the rationale for testing entry pathways, as laid out in the context of MxB, to be confusing. As noted, the authors have already published that VSV-G is more sensitive to IFITM proteins. The data seem to indicate that the MxB restriction is 5 to 10-fold for either context, so that it was unclear to me how the entry pathway was relevant to MxB restriction.

6) For Figure 4E results, for a general audience, the authors need to define in the text what Vpu does (e.g., subsection “TRIM5alpha, IFITM1 and Tetherin are additional ISGs that contribute to the IFN block”, last paragraph).

7) One caveat is that in no instances were the KO gene activities rescued to confirm KO specificity. The authors might comment on why this was unnecessary.

8) The authors mention "genetic conflicts" on several occasions as a possible explanation for why the virus is unable to completely antagonize human RFs such as TRIM5alpha and BST2, without always explaining this concept very clearly. For example, a major conclusion is that T5alpha, etc. affects "well-adapted" HIV-1-LAI (Discussion, first paragraph), but what does "well-adapted" mean? They mention "conflicting evolutionary pressures" acting on the genome but only much later in the Discussion do they begin to explain what they mean by this (e.g., that evasion might come at a cost for other viral functions). They also mention "concurrently-acting RF barriers" as a potential explanation but this was also unclear to me.

9) One suggestion for the Discussion would be to address the question of how much ZAP might limit the application of this screening strategy for primary cells wherein it would be difficult and perhaps less-than-ideal to inactivate ZAP directly. Instead, would it be feasible to remove CpG dinucleotides from the library? As noted above, I thought there could also be more emphasis on the data that nicely implicates both TRIM25 and N4BP1 as downstream regulators of ZAP restriction.

10) The comparison of this screen to that of Park et al. should be a separate paragraph in the Discussion, and more could be said here, both to better explain the Park screen and better articulate the pros and cons of either screen and their findings. Would be useful for readers.

11) APOBECs are brought up in the Discussion and are apparently not detected in this screen due to its "single-round" nature. Would it be possible to modify the screen wherein it could detect APOBEC-like activities (i.e., impacting RT, integrase, or some other aspect of multi-round replication)? This could be discussed.

Text edits:

1) Much of the paper is written in present tense.

2). For a broad audience, be sure to define abbreviations and jargon in Materials and methods and elsewhere, and also other esoteric terms (e.g., "massively-parallel").

3) Introduction, second paragraph; might cite original papers or more recent reviews, for example MxB not reported until 2013.

4) "MxB is a the"; "a" vs. "the" will make a big difference in this sentence.

5) "these pseudotyped", rather "of VSV-G pseudotyped"?

6) "may due".

7 Subsections “Cell Culture”, “Virus and lentivirus production”, and “Knockout and Knockdown Cell Pools and Clones”, some mis-capitalizations.

Additional data files and statistical comments:

Rigor and statistics seemed adequate.

*Reviewer #2:*

The MS from OhAinle and colleagues describes a co-packaging lentiviral vector-based CRISPR screen using a bespoke 1900 gene library to identify interferon-regulated antiviral factors that target HIV-1 LAI, or in parallel, genes enriched in HIV-1 target cells that act as putative cofactors. Because the CRISPR vector can be mobilized by replication competent HIV-1 replication in the same cell, sgRNA-CAS9 vectors targeting putative antiviral genes will be overrepresented in the supernatant of infected cells pretreated with IFN. Conversely, putative cofactors will be underrepresented under control conditions. In-built into the screen were key controls that should be screened out under both conditions (HIV entry factors and those involved in the transduction of IFN signalling), which, importantly, they were. The screen revealed a number of ISGs previously shown to inhibit retroviral replication. Of these, ZAP, TRIM25 and N4BP1 appear to be targeting the CRISPR vector rather than HIV itself, most likely because of the high CpG content of the CAS9-P2A-PuroR mRNA/vector genome RNA. For the other ISGs identified, the authors show that hTRIM5, tetherin, MxB and several IFITMs contribute to the IFN-induced inhibition of HIV replication. The relative sensitivity of HIV-1 to different IFITMs vs MxB depends largely on whether viral entry is mediated via VSV-G or the native CXCR4-using LAI envelope. Finally, by examining sgRNAs that are under-represented, they identified SEC62 and TLR2 as putative cofactors in THP-1 cells.

This an interesting and well-written study. The screen itself is elegant, and very well controlled, and as such represents a useful resource for the field and a powerful strategy to identify novel HIV-associated cellular factors. In THP-1, the screen has not identified any novel HIV-1 restriction factors. This maybe because there are no new major ones to be found, but there are also some limitations in the screen as presented. Moreover, while the authors have confirmed the importance of MxB and tetherin in the IFN-induced resistance to HIV-1 replication, novel aspects of some of the other factors could be developed further.

I would encourage the authors to consider some of the following:

The screen in THP-1 is limited to X4-using or VSV-G pseudotyped HIV-1 for which they use the lab-adapted prototypic strain LAI. It would be interesting to see whether the same genes are identified if a primary HIV-1 isolate was used instead. This would either require challenging the cells with a VSV-G pseudotype or making the THP-1 ZAP-KO cells CCR5+ve. The authors touch on the idea that the lab-adapted strain may be have residual sensitivity to certain ISGs, and there is good evidence of a spectrum of IFN sensitivities amongst primary HIV isolates. The system the authors have developed gives them the opportunity to do interesting comparative studies which will yield further novel observations. I would encourage the authors at least to discuss this more fully if they feel re-running the screen with a different virus/virus mutant is beyond the scope.

The relative impact of MxB vs. IFITM restrictions depending on route of entry would be strengthened by comparing other HIV-1 envelopes, given other publications in the field. Of note, a very closely related molecular clone derived from NL/HxB, but not an Env-deleted VSV pseudoype of the same virus, has been shown to be completely insensitive to IFNa in the THP-1 cells despite robust Mx2 induction (Rihn et al., JVI 2017).

I strongly suspect that the identification of tetherin in the screen is because the Vpu encoded LAI/IIIB lineage is a sub-optimal tetherin antagonist. While the authors allude to compromised Vpu function in the Discussion, they should refer to Pickering et al., 2014, where essentially the same Vpu allele (NL4.3) was specifically shown to be an inferior tetherin antagonist compared to most primary isolates tested.

The identification of human TRIM5 is somewhat surprising considering that HIV-1 is supposed to be largely resistant to it. How much does TRIM5 contribute to the enhanced IFN-sensitivity of certain capsid mutants? Again, is TRIM5 contributing to IFN sensitivity of other HIV-1 isolates in the same system. Likewise, there is preliminary evidence that N4BP1 could be a ZAP cofactor, but this is not further characterized.

The identified novel dependency factors could be characterized more fully. The identification of TLR2 and (to a lesser extent) its adaptor MyD88 as dependency factors suggests downstream signalling. Can this phenotype be explained by knockout affecting NFkB activity and therefore the activity of the viral LTR (of note NFkB1 is also a hit)? As there is no TLR2 ligand added to the culture, what could be stimulating the system here? SEC62 appears to be affecting CD4, but not CXCR4, surface expression and this correlates with sensitivity of LAIwt but not LAI(VSV). Are other HIV-1 envelopes that require different cell densities if CD4 more or less sensitive to SEC62 knockout?

*Reviewer #3:*

This manuscript from Ohainle and colleagues describes an elegant CRISPR/deep sequencing-based screen that was designed to identify interferon-stimulated genes that affect early or late stages of HIV infection. The results are clear and significant and identify type 1 IFN pathway genes (a form of positive control), restriction factors and dependency factors that enhance virus replication. In addition, the effects of ZAP registered in the assay owing to CG-high vector sequences, and the screen was modified accordingly. The top four restriction factors were MxB, IFITM1, tetherin and TRIM5alpha, with UBE2L6, LGALS3BP and SAMD9L scoring weakly. A number of dependency factors were identified and, given that the sgRNA library was not whole genome, this suggests that many more could be identified by applying this experimental approach.

The screen itself has been cleverly designed, and it is reassuring that many of the expected genes crop up as hits. Though not a criticism of the study, one may question the extent of new HIV biology that has been learned? For instance, the identification of human TRIM5alpha in this screen is unexpected, but no attempt was made to clarify why it was identified yet has largely been regarded by the field as an ineffective inhibitor of HIV-1? It is suggested that this could be due to its induced expression as an ISG, but this was not addressed. Similarly, TRIM5alpha restriction was bypassed by VSVg pseudotyping (Figure 3E): is this a general feature of TRIM5 restriction, or is it a characteristic of this experimental set-up and/or THP cells?

Did the authors undertake a deep sequencing analysis of the PIKA-HIV library prior to generating the transducing virus stock in 293T cells? It would be useful to know if the transduced THP population already carried some bias in its sgRNA content (for instance, it might be anticipated that guides recognising dependency factors affecting the late stage of virus replication would be under represented).

Do the authors have any theories regarding TLR2 as a dependency factor? Do they believe this effect is via signalling, and if it is, what the agonist may be?

A number of the knock-out populations should be validated either by western blot or endonuclease assay.

---

## [Author Response]

[…] Overall, this is a fascinating project, very nicely performed, and the majority of paper is clearly laid out and well-written. While the major contribution is the derivation and optimization of a powerful new screening method for identifying HIV-host interactions, validation of the screen also yields interesting new restriction and host dependency factors impacting the HIV life cycle and insights regarding the multi-layer nature of cellular antiviral immunity. A particularly intriguing finding is the demonstration that human versions of antiviral host proteins typically thought to be circumvented by HIV-1 (e.g., TRIM5a and BST2/Tetherin) retain a significant level of anti-HIV activity in IFN-treated THP-1 cells. However, the new HIV biology could be enhanced by increased focus on their mechanisms, at least in the Discussion (e.g., the identification of human TRIM5alpha in this screen was unexpected, but no attempt was made to clarify how it functions, which is of interest given that it has largely been regarded by the field as an ineffective inhibitor of HIV-1). Nevertheless, most of the paper's weaknesses are minor and largely pertain to clarifications that could be made to the description and interpretation of a subset of experiments.

We thank the reviewers for their support of this study. As described below, the manuscript has been considerably strengthened and expanded by repeating the entire CRISPR screen with an R5-tropic primary strain of HIV. This new data (as well as a repeated VSV-G pseudotyped HIV screen) allows us to make additional comparisons between viral strains. We also have added data directly to the figures about the validation of all of the knockouts described in the manuscript.

Major Issue:At least a subset of the knock-out populations should be validated experimentally (e.g., by western blot, endonuclease or sequencing assays). Similarly, in no instances were the KO gene activities rescued to confirm KO specificity. The authors might at least comment on why they felt this was unnecessary.

We did validate all of the individual knockout data in this paper and had put most of this data in the Supplementary Data (originally Supplementary Table 6). However, since all three reviewers missed this information, it was obviously poorly referred to in the text. We now specifically include in the figures the validation of the knockouts by sequencing assays, Western blotting or Flow Cytometry. This data has been added to Figure 2, Figure 4, and Figure 5 as well as Figure 4—source data 1.

Minor Issues:1) In Figure 1, the rationale for repairing the LTR to generate packageable RNA species could be more clearly described for a general audience. Also, is there potential for promoter interference in this context?

We agree that the significance of repairing the 3’ LTR on the screen strategy should be better described for a general audience. We have added text to better describe why this is critical in the subsection “An ISG-specific knockout screen that packages sgRNA-encoding lentiviral genomes into HIV virions”. While there is a potential for promoter interference in the vector, it is not sufficient to prevent the screen from working.

2) Why was cellular gDNA was sequenced and not cell-associated RNA? While gDNA would give a representation of each guide RNA as integrated into the cell pool, it does not confirm that the guide RNAs are expressed. Could this cloud the analysis? A related question is whether essential genes that may be involved in the viral life cycle can be detected in this screen based on loss of sequences? Finally, how was ZAP still identified as a mid-level hit even in ZAP KO cells (subsection “An iterative PIKAHIV screen in ZAP-KO cells identifies a panel of ISGs that inhibit HIV in THP-1 cells”, second paragraph)?

Sequencing of the cell-associated RNA is a reasonable idea. However, this is complicated by the fact that guide RNAs are expressed from the pol III promoter in the vector but these sequences are also present in the full-length HIV-CRISPR RNA produced from transcription off the LTR. Thus, we could not easily distinguish cellular RNA that contained in the viral genome versus the subgenomic RNA that would lead to knockout. In addition, LTR-driven expression could be affected by gene knockouts. The library was introduced into the THP cells at 500X coverage. Therefore, differences in expression levels between integration sites should be averaged out. We therefore believe that sequencing of integrated constructs in the genomic DNA as a read out of the frequency of each guide RNA that is present in the cell population serves as a sufficient denominator for comparing the enrichment or depletion of HIV-CRISPR viral genomes in the supernatant.

Yes, the reviewer is correct that genes essential for the cell would likely not be read out in this screen. However, as we chose a set of interferon-induced genes, we would expect essential genes to be under-represented in the library. In fact, we are only missing 9 out of 15,348 sgRNAs on sequencing the genomic DNA collected at the time of viral supernatant collection (we have added this information to the Materials and methods).

ZAP is not identified as a hit in the screen in the ZAP-KO cells. It is ranked #1647 out of 3812 genes similar to the non-targeting controls (NTCs). This is the background of the assay.

3) N4BP1 could be highlighted more strongly (e.g., in Abstract) considering that it appears to be a new potential regulator of the ZAP pathway.

Thank you for pointing this out. The data that N4BP1 co-segregates in the HIV-CRISPR screen with ZAP is shown in the manuscript. However, we have now added additional data in the current manuscript that shows that N4BP1 is not an essential co-factor for ZAP inhibition of alphaviruses (Figure 2—figure supplement 1E). Thus, our data suggests that ZAP may have different requirements for inhibition of different targets. However, our attempts at further determining the mechanism of N4BP1 mechanism have not yielded satisfactory answer. As we do not have further data to support a model of how this might occur we prefer not to highlight this aspect of the study in the Abstract.

4) For Figure 2C, it could be better explained why it makes sense that gene correlations are stronger for the ZAP-KO cells than for wild-type cells.

We have added text to further explain the logic of why the gene correlations should improve in the absence of ZAP (which our data suggests inhibits our vector, but not HIV itself).

5) For Figure 3 results, please clarify the rationale for testing entry pathways (as laid out in the context of MxB). As noted, the authors have already published that VSV-G is more sensitive to IFITM proteins. The data seem to indicate that the MxB restriction is 5 to 10-fold for either context, so it is unclear how the entry pathway was relevant to MxB restriction.

We apologize for not explaining the logic better. A previous paper suggested that MxB is not a restriction factor for HIV (Opp et al., 2015). However, the authors of that study measured restriction of VSV-G pseudotyped HIV in contrast to wildtype HIV infections. Our results now provide an explanation for the lack of detectable restriction by MxB as there is a more dominant inhibitor of the VSV-G pseudotype, namely IFITM3, which masks the effect of MxB. We now state this more clearly in the Results. Further we have replaced the VSV-G pseudotype screen data with a newer VSV-G pseudotype screen performed in ZAP-KO THP-1 cells that more clearly shows these phenomena (Figure 3E and Figure 6—figure supplement 1: the figure legend, Results and Materials and methods have been updated accordingly). In addition, a recent paper that comes to the same conclusion that the degree of the IFN response due to MXB depends on the mode of viral entry is now cited (Xu, Pan, and Liang, 2018).

6) For Figure 4E results, for a general audience, the authors need to define in the text what Vpu does (e.g., subsection “TRIM5alpha, IFITM1 and Tetherin are additional ISGs that contribute to the IFN block”, last paragraph).

We have now added this information in the text.

7) The authors mention "genetic conflicts" on several occasions as a possible explanation for why the virus is unable to completely antagonize human RFs such as TRIM5alpha and BST2, without always explaining this concept very clearly. For example, a major conclusion is that T5alpha, etc. affects "well-adapted" HIV-1-LAI (Discussion, first paragraph), but what does "well-adapted" mean? They mention "conflicting evolutionary pressures" acting on the genome but only much later in the Discussion do they begin to explain what they mean by this (e.g., that evasion might come at a cost for other viral functions). They also mention "concurrently-acting RF barriers", but this potential explanation should be clarified.

We agree that these concepts were not sufficiently explained. We now expand the Discussion to better explain what we mean by these genetic conflicts, conflicting evolutionary pressures and concurrently-acting RF barriers. We have moved up our discussion on this to come before discussion of more technical aspects of the screen. By “well-adapted”, we mean an HIV strain that was isolated from a person with chronic infection and high viral load.

8) For the Discussion, we suggest addressing the question of how much ZAP might limit the application of this screening strategy for primary cells wherein it would be difficult and perhaps less-than-ideal to inactivate ZAP directly. Instead, would it be feasible to remove CpG dinucleotides from the library? As noted above, there could also be more emphasis on the data that nicely implicates both TRIM25 and N4BP1 as downstream regulators of ZAP restriction.

We have been trying very hard to adapt this screen to primary cells. Besides the CpG problem, we have identified other technical problems that need to be resolved before it is successful. We prefer not the address this extensively in the Discussion until we have more answers.

9) The comparison of this screen to that of Park et al. should be a separate paragraph in the Discussion, and more could be said here, both to explain the Park screen better and articulate the pros and cons of either screen and their findings.

We have expanded this part of the Discussion to better compare the Park et al. screen with our screen (both the methodology and the results).

10) APOBECs are brought up in the Discussion and are apparently not detected in this screen due to its "single-round" nature. Would it be possible to modify the screen wherein it could detect APOBEC-like activities (i.e., impacting RT, integrase, or some other aspect of multi-round replication)? This could be discussed.

We are working on this, but prefer not to expand the discussion on this extension of HIV-CRISPR screening until we have some data to support this additional approach.

11) The screen in THP-1 is limited to X4-using or VSV-G pseudotyped HIV-1, using the lab-adapted prototypic strain LAI. It would be interesting to see whether the same genes are identified if a primary HIV-1 isolate was used instead. This would either require challenging the cells with a VSV-G pseudotype or making the THP-1 ZAP-KO cells CCR5 positive. The authors touch on the idea that the lab-adapted strain may be have residual sensitivity to certain ISGs, and there is good evidence of a spectrum of IFN sensitivities amongst primary HIV isolates. The system the authors have developed gives them the opportunity to do interesting comparative studies which will yield further novel observations. The authors should at least discuss this more fully if they feel re-running the screen with a different virus/virus mutant is beyond the scope.

We agree that interesting comparisons can be made by screening more viruses and have now repeated PIKA_HIV_ screening with an R5-tropic virus derived from a primary isolate. We introduced CCR5 into the THP-1 ZAP-KO cells, introduced the PIKA library into the cells, and then challenged them with HIV-_1Q23.BG505_, a clade A R5-tropic molecular clone (the Q23 molecular clone with envelope from BG505). This data is now shown in a new Figure 6 (6A, 6B and 6C). A comparison of the screens now forms a major part of the Discussion. In short, the major conclusions of important factors are in many ways similar for all three viruses, but there are also some interesting differences that we can now highlight.

In addition, we have now challenged these same ZAP-KO PIKA_HIV_ THP-1 cells with the VSV-G pseudotyped HIV-1_LAI_ virus and are now able to detect a set of significantly-scoring HIV dependency factors in the VSV-G screen. This data is shown in Figure 6—figure supplement 1. As above, the comparison of the results of these additional screens now serves as major basis of the Discussion.

12) Similarly, the relative impact of MxB vs. IFITM restrictions depending on route of entry would be strengthened by comparing other HIV-1 envelopes, given other publications in the field. Of note, a very closely related molecular clone derived from NL/HxB, but not an Env-deleted VSV pseudoype of the same virus, has been shown to be completely insensitive to IFNa in the THP-1 cells despite robust Mx2 induction (Rihn et al., JVI 2017).

As described to point 11, we have now repeated the entire screen with both a recombinant of R5-tropic primary isolates and in the ZAP-KO cells with VSV-pseudotyped HIV-1_LAI_ virus. These new screens now more completely explain the relative impacts of MXB and IFITM on restriction that is dependent on the mode of entry.

13) Note that the identification of tetherin in the screen could be because the Vpu encoded LAI/IIIB lineage is a sub-optimal tetherin antagonist. While the authors allude to compromised Vpu function in the Discussion, they should refer to Pickering et al., 2014, where essentially the same Vpu allele (NL4.3) was specifically shown to be an inferior tetherin antagonist compared to most primary isolates tested.

Thank you for bringing up this point. In our new screen with an R5-tropic virus (HIV-1_Q23/BG505_), Tetherin is now much further down on the list and does not appear to be a restriction factor. Therefore, the reviewer is likely correct that the Vpu of HIV-1_LAI_ is sub-optimal. This data is entirely consistent with the hypothesis that viruses from chronic infections become more interferon-sensitive—possibly because of differences in potency between the viral antagonists. We now highlight this difference in the screens and also include the reference pointed out by the reviewer.

14) As noted above, the manuscript would benefit from additional mechanistic information (or at least increased discussion) for several of the more interesting "hits". In particular, identification of human TRIM5 is somewhat surprising considering that HIV-1 is supposed to be largely resistant. How much does TRIM5 contribute to the enhanced IFN-sensitivity of certain capsid mutants? Is TRIM5 contributing to IFN sensitivity of other HIV-1 isolates in the same system? It is suggested that the TRIM5 effects could reflect induced expression as an ISG, but this was not addressed. Similarly, TRIM5alpha restriction was bypassed by VSVg pseudotyping (Figure 3E): is this a general feature of TRIM5 restriction, or is it a characteristic of this experimental set-up and/or THP cells? Likewise, there is preliminary evidence that N4BP1 could be a ZAP cofactor, but this is not further characterized. Finally, the identified novel dependency factors could be characterized more fully. The identification of TLR2 and (to a lesser extent) its adaptor MyD88 as dependency factors suggests downstream signaling. Do the authors have any theories regarding TLR2 as a dependency factor? Do they believe this effect is via signaling, and if it is, what the agonist may be? Can this phenotype be explained by knockout affecting NFkB activity and therefore the activity of the viral LTR (of note NFkB1 is also a hit)? As there is no TLR2 ligand added to the culture, what could be stimulating the system here? SEC62 appears to be affecting CD4, but not CXCR4, surface expression and this correlates with sensitivity of LAIwt but not LAI(VSV). Are other HIV-1 envelopes that require different cell densities if CD4 more or less sensitive to SEC62 knockout?

These are all interesting points. We now show TRIM5 to also be a hit in the additional screen using the R5-tropic clade A HIV-1_Q23/BG505_ virus. As we show in the paper, the effect of TRIM5 in IFN-treated THP cells is still relatively modest in the knockout cells—about 4-fold IFN inhibition as compared to 7-fold in control cells (compare NTC with TRIM5_1 and TRIM5_2 in Figure 4B and 4C). While this is much less than the amount of restriction of HIV by over-expression of the rhesus TRIM5, careful examination of published results (including our own), substantiate small effects of human TRIM5 on HIV-1 (reproducible across different labs using different methods). Further we now show TRIM5 inhibition of the VSV-G pseudotyped virus screened in the ZAP-KO THP-1 cells (Figure 3E), highlighting its ability to inhibit HIV entering via different envelopes. TRIM5 is induced by IFN (Figure 2E and Figure 2—source data 1), so yes, its effect would be expected to be greater in the presence of IFN. We are in the process of screening CA mutants, but believe that data better fits in a different manuscript. A reference (van Manen et al., 2008) suggesting a correlation of TRIM5 polymorphisms with HIV clinical course has been added.

As described above, we now add data that N4BP1 is not a co-factor for ZAP’s role in inhibiting alphaviruses (Figure 2—figure supplement 1E). However, despite our considerable efforts, we do not have further mechanistic data on the role of N4BP1 in inhibition of the viral vector. As the emphasis of this manuscript is on factors that inhibit HIV itself, we would prefer to leave that work to others.

We do not have further data on the role of TLR2 and MyD88 as dependency factors. However, as the reviewer points out NFKB1 is also a dependency factor, so it is possible that signaling through NFKB1 is occurring in the absence of added exogenous ligand. Further we did find mycoplasma in some THP-1 cultures tested after the completion of experiments which could be serving as a TLR2 agonist (Heidegger et al., 2015). There is also evidence that HIV infection itself can agonize TLR2 (Henrick, Yao, Rosenthal, and team, 2015). We now cite more papers that support this interpretation in the Discussion but do not have further data on this point.

SEC62 comes up as a dependency factor for both the HIV-1_LAI_ (X4-tropic) and HIV-1_Q23/BG505_ (R5-tropic) viruses we’ve screened, but does not come up as a dependency factor in the VSV-G pseudotype virus screen. This data supports the interpretation that the target of SEC62 is CD4 as the requirement for CD4 is shared by both the X4- and R5-tropic viruses but not VSV/G-pseudotyped virus. Further the two wild type HIV viruses we’ve tested likely require different CD4 levels to infect cells (Humes and Overbaugh, 2011), suggesting that the effect of SEC62 is independent of efficiency of CD4 usage. It is formally possible that the effect of SEC62 on CD4 correlates with infection efficiency but is not causative.

15) Did the authors undertake a deep sequencing analysis of the PIKA-HIV library prior to generating the transducing virus stock in 293T cells? It would be useful to know if the transduced THP population already carried some bias in its sgRNA content (for instance, it might be anticipated that guides recognizing dependency factors affecting the late stage of virus replication would be under represented).

Yes, we did sequence the PIKA-HIV library prior to transduction. We now add this data as Figure 1—figure supplement 2.

Text edits:1) Much of the paper is written in present tense.

Corrected.

2) For a broad audience, be sure to define abbreviations and jargon in Materials and methods and elsewhere, and also other esoteric terms (e.g., "massively-parallel").

We have worked on this and removed jargon and obscure abbreviations.

3) Introduction, second paragraph; might cite original papers or more recent reviews, for example MxB not reported until 2013.

We cite the original papers describing the role of MXB on HIV infection and a recent review.

4) "MxB is a the"; "a" vs. "the" will make a big difference in this sentence.

Corrected.

5) "these pseudotyped", rather "of VSV-G pseudotyped"?

Corrected.

6) "may due".

Corrected.

7) Subsections “Cell Culture”, “Virus and lentivirus production”, and “Knockout and Knockdown Cell Pools and Clones”, some mis-capitalizations.

Corrected.